# Galectin-9 suppresses B cell receptor signaling and is regulated by I-branching of N-glycans

N. Giovannone [1,2], J. Liang[1], A. Antonopoulos[3], J. Geddes Sweeney[1,2], S. L. King[1], S.M. Pochebit[2,4], N. Bhattacharyya[5,6], G.S. Lee[6], A. Dell[3], H.R. Widlund[1], S.M. Haslam [3] & C.J. Dimitroff[1,2]

Leukocytes are coated with a layer of heterogeneous carbohydrates (glycans) that modulate immune function, in part by governing specific interactions with glycan-binding proteins (lectins). Although nearly all membrane proteins bear glycans, the identity and function of most of these sugars on leukocytes remain unexplored. Here, we characterize the N-glycan repertoire (N-glycome) of human tonsillar B cells. We observe that naive and memory B cells express an N-glycan repertoire conferring strong binding to the immunoregulatory lectin galectin-9 (Gal-9). Germinal center B cells, by contrast, show sharply diminished binding to Gal-9 due to upregulation of I-branched N-glycans, catalyzed by the β1,6-*N*-acetylglucosaminyltransferase GCNT2. Functionally, we find that Gal-9 is autologously produced by naive B cells, binds CD45, suppresses calcium signaling via a Lyn-CD22-SHP-1 dependent mechanism, and blunts B cell activation. Thus, our findings suggest Gal-9 intrinsically regulates B cell activation and may differentially modulate BCR signaling at steady state and within germinal centers.

[1] Department of Dermatology, Brigham and Women's Hospital, Boston, MA 02115, USA. [2] Harvard Medical School, Boston, MA 02115, USA. [3] Department of Life Sciences, Imperial College London, London SW7 2AZ, UK. [4] Department of Pathology, Brigham and Women's Hospital, Boston, MA 02115, USA. [5] Department of Surgery, Division of Otolaryngology, Brigham and Women's Hospital, Boston, MA 02115, USA. [6] Department of Otology and Laryngology, Harvard Medical School, Boston, MA 02115, USA. Correspondence and requests for materials should be addressed to S.M.H. (email: s.haslam@imperial.ac.uk) or to C.J.D. (email: cdimitroff@bwh.harvard.edu)

Antibody (Ab)-mediated immune responses require the coordinated activation and differentiation of antigen inexperienced (naive) B cells into Ab secreting cells (plasma cells) and B cells with immunological memory (memory B cells). The most effective Abs arise from the germinal center (GC) reaction, in which B cells undergo Ab affinity maturation, acquire specialized effector functions via isotype switching, and develop into long-lived memory and plasma cells[1]. These processes are crucial for pathogen elimination and the development of durable immunity; yet, when aberrantly targeted against host, graft, or innocuous antigen, these same processes can drive autoimmune and autoinflammatory disease, transplant rejection, and chronic allergy. Consequently, deciphering the molecular mechanisms governing B cell activation and differentiation remains a central goal in understanding both normal and aberrant immune responses[2].

Glycoprotein-linked carbohydrates, or glycans, are increasingly being recognized for their key regulatory functions during immune homeostasis and inflammation[3,4]. Through post-translational addition to either asparagine (N-linked) or serine/threonine resides (O-linked), glycans can alter the biophysical properties of their protein scaffolds[3,4]. Alternatively, in a specific manner, glycan can themselves serve as binding determinants for a vast array of endogenous glycan binding proteins (lectins) that have myriad functions in the immune system, such as regulation of leukocyte homing, adhesion, pathogen sensing, and immune signaling[3,4]. As with traditional receptor–ligand interactions, the upregulation, downregulation, or modification of glycan ligands has the ability to regulate lectin binding and activity[3–5].

In B cells, among the best understood roles for lectin–glycan interactions in humoral immune responses are those of sialoglycans and sialic acid-binding lectins (siglecs) in B cell receptor (BCR) signaling. Myriad reports have implicated both CD22 (Siglec 2) and Siglec 10 (Siglec G in mice) lectins as inhibitory co-receptors of BCR signaling that modulate B cell activation and peripheral tolerance[6]. Mice deficient for CD22[7], Siglec G[8], or both[9] exhibit varying degrees of autoimmunity, and congenital mutations in an enzyme modifying CD22 sialic acid ligands are associated with increased risk of autoimmune disease in humans[6,10]. However, while pivotal advances have been made in understanding the contribution of sialic acid/siglec interactions to B cell function, sialic acids represent only a small fraction of the glycan moieties normally expressed by a mammalian cell[3]; the vast majority of glycans expressed by B cells have remained largely undefined. Just as characterizing the global gene expression patterns of immune subsets has yielded critical insights into the function of disparate immune populations, including B cells[11], defining the global glycan repertoire (the glycome) of immune cells will likely uncover novel lectin–glycan interactions regulating immunity[12].

Here, we profile the global N-glycosylation repertoire of B cells at several stages of differentiation. From this analysis, we uncover a role for the immunomodulatory lectin galectin-9 (Gal-9) as a B cell-intrinsic regulator of BCR signaling and activation whose binding is differentially regulated by I-branching of N-glycans. Our data highlight the potential for glycomics-driven approaches to identify novel regulators of adaptive immunity.

## Results

**B cells differentially express I-branched poly-N-acetyllactosamine.** GCs are dynamic anatomical niches where B cells rapidly proliferate, undergo Ab affinity maturation, and differentiate to memory B cells and plasma cells that confer long-lived immunity[1]. Accordingly, these cellular transitions require significant reprogramming at each stage in order to drive new B cell

functionality[1,11]. We hypothesized that B cell reprogramming during GC responses may extend to N-glycans, which, depending on N-glycan expression patterns, may confer or block interactions with specific endogenous lectins.

To test this hypothesis, we took advantage of an emerging and powerful tool for glycan analysis, N-glycome mass spectrometry (MS), which permits highly sensitive interrogation of global N-glycan structures present on membrane glycoproteins[12,13]. Specifically, we FACS-sorted human naive, GC, and memory B cells from lymphoid tissue (human tonsil) to high purity (>95%), liberated N-glycans by enzymatic cleavage with PNGase F, and performed matrix-assisted laser desorption/ionization time of flight (MALDI-TOF) MS on the released N-glycans (Fig. 1a–e, Supplementary Fig. 1, Supplementary Fig. 2a–d). We found that all three B cell N-glycomes contained strikingly high levels of N-acetyllactosamines (LacNAc) appearing as chains of 2–4 units, termed poly-N-acetyllactosamine (poly-LacNAc), on multiple N-glycan antennae (Fig. 1c, d). We noted similar findings by flow cytometric staining of B cells with several complex N-glycan- and LacNAc-specific plant lectins, including *Erythrina cristagalli* (ECA), *Solanum tuberosum* (STA), *Lycopersicon esculentum* (LEA), and *Phaseolus vulgaris* (PHA-L) lectins (Supplementary Fig. 3a, b).

Deeper analysis by tandem MS revealed important structural differences between poly-LacNAcs on naive, GC, and memory B cells: while naive and memory B cell poly-LacNAcs were composed of 2–4 LacNAc units arranged in a straight chain (linear poly-LacNAc), GC B cell poly-LacNAcs were slightly shorter (maximum of 3 units) and branched by additional LacNAcs in an arrangement known as I-branches (also called "adult I" blood group antigen) (Fig. 1c–e, Supplementary Fig. 2a–d). Consistent with expression of I-branched poly-LacNAcs[14], GC B cells showed exceptionally high levels of binding to LEA and STA plant lectins, despite similar or slightly decreased expression of complex N-glycans and terminal LacNAcs (Supplementary Figure 3a, c). Moreover, immunohistochemical staining of tonsil tissue with STA lectin revealed diffuse staining in GC compared to mantle zones (Supplementary Fig. 3d). Strong punctate STA staining scattered through GCs was also apparent, possibly corresponding with tingible body macrophages, although with unclear significance.

Taken together, these data demonstrate that the B cell N-glycome is characterized by complex, poly-LacNAc-rich N-glycans that are predominantly linear in naive and memory B cells, but modified with I-branches at the GC stage.

**Naive and memory B cells, but not GC B cells, bind Gal-9.** Poly-LacNAc containing multi-antennary N-glycans are known to be canonical binding determinants for galectins[15,16]. Galectins, also called S-type lectins, have broad expression in both immune and stromal tissues and perform a constellation of immunoregulatory functions through binding to an array of glycosylated receptors[15–22]. In particular, Gal-9 is known to have potent regulatory effects on adaptive immunity, including dampening of inflammatory T cell responses via binding to T cell immunoglobulin and mucin-domain 3 (TIM-3)[17–22], and has been documented to have strong binding affinity for poly-LacNAcs[16,22]. In B cells, Gal-9 deficient mice are reported to have increased B cell proliferation, enlarged GCs, and stronger Ab responses to infection, and Gal-9 treatment has been observed to inhibit vaccination-induced antibody responses and ameliorate pathology in mouse models of systemic lupus erythematosus[17–20,23]. Yet, a direct mechanism of action of Gal-9 on B cells has remained unclear. Given robust expression of Gal-9-binding glycans by B cells (Fig. 1c–d), we sought to test whether Gal-9

may directly bind and regulate B cells in a glycan-dependent manner.

To this end, we assessed Gal-9 binding to naive, GC, and memory B cells ex vivo by flow cytometry. Consistent with their expression of linear poly-LacNAc-containing N-glycans, naive and memory B cells showed strong binding to Gal-9 that was

glycan-dependent, as evidenced by absence of binding in the presence of lactose, a competitive inhibitor of galectin carbohydrate-binding activity (Fig. 2a, top; lactose, gray histogram). Strikingly, however, in comparison to the strong binding of Gal-9 to naive and memory B cells, GC B cells showed substantially diminished binding that inversely correlated with

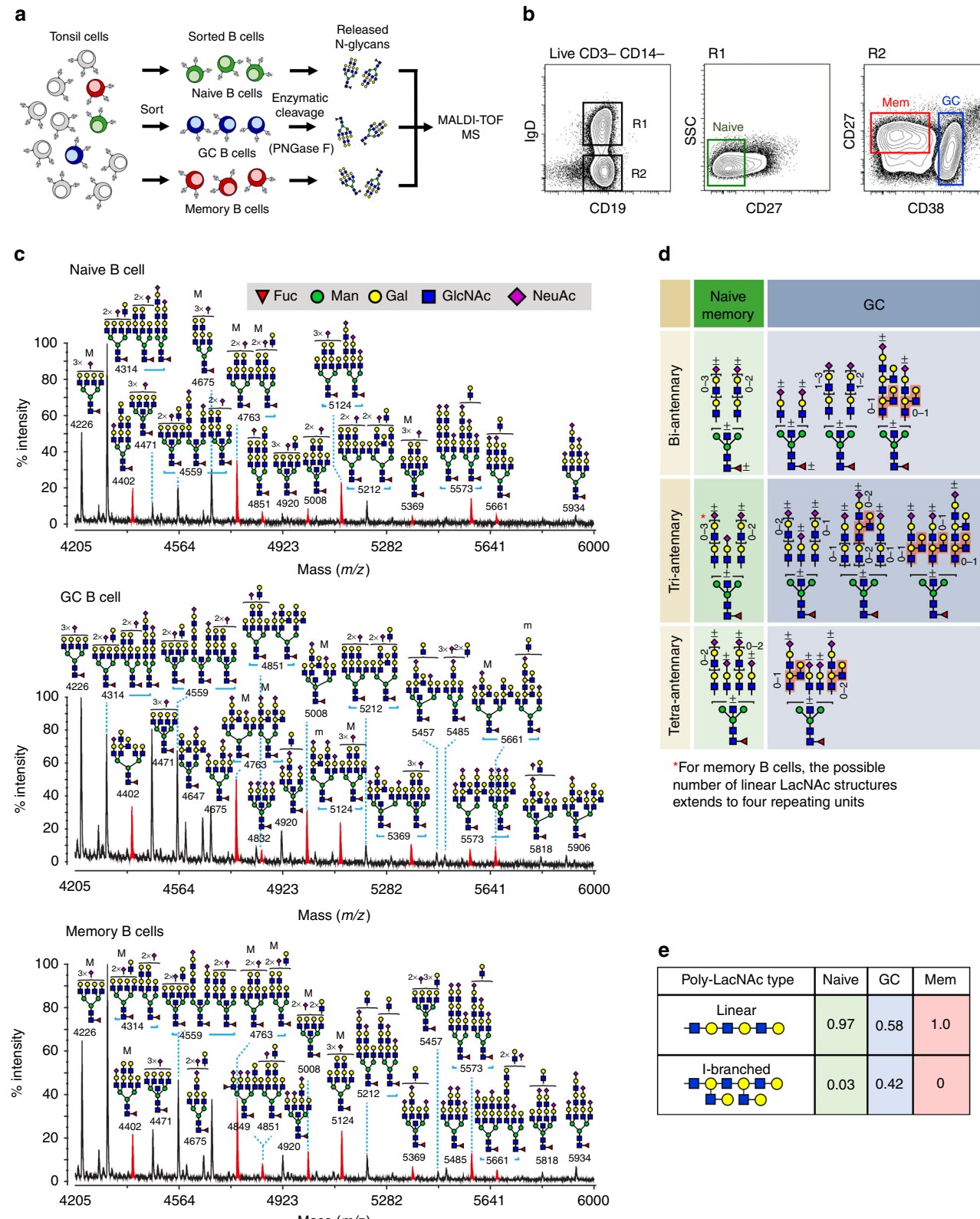

I-branch expression (Fig. 2a). By contrast, GC B cell binding to another galectin family member, Gal-1, was only minimally impacted, suggesting that the loss of binding may be Gal-9 specific (Fig. 2a). We observed similar binding differences over a range of Gal-9 staining concentrations (Supplementary Fig. 4a). Collectively, these data suggested Gal-9 binding may be differentially regulated between naive, memory, and GC B cells by global alterations in N-glycosylation.

**I-branching by GCNT2 physiologically regulates Gal-9 binding.** The striking inverse relationship between expression of I-branched poly-LacNAc and Gal-9 binding in GC B cells led us to hypothesize that I-branches directly modulate Gal-9 binding. However, because GC B cells also exhibited slightly reduced expression of complex N-glycans (Supplementary Fig. 3c) and somewhat shorter LacNAc chains (Fig. 1c, d), each of which could theoretically contribute to reduced Gal-9 binding, we sought to directly test the effect of I-branching on Gal-9 binding activity.

On N-glycans, I-branching is thought to be exclusively mediated by the $\beta 1,6$-$N$-acetylglucosaminyltransferase GCNT2 (formerly IGNT) (Fig. 2b)[24,25]. In line with this, we found GCNT2 expression to be significantly increased in GC B cells compared to naive and memory B cells (Fig. 2c). Therefore, to directly assay the effect of I-branching on Gal-9 binding, we generated stable knockdown and overexpression variants of GCNT2 in a GC-derived cell line possessing high GCNT2 (Ramos B cells) and a non-GC derived B cell line with low GCNT2 (NUDUL-1 B cells), respectively (Supplementary Fig. 4b, c). As expected, manipulation of GCNT2 expression in B cells led to corresponding alterations in I-branch expression while having minimal or negligible effect on complex N-glycan expression, sialylation, or total LacNAc content (Supplementary Fig. 4d-f). Remarkably, B cell overexpression of GCNT2 resulted in a near-complete loss of Gal-9 binding (Fig. 2d, top). By contrast, GCNT2 knockdown resulted in over fivefold enhanced Gal-9 binding (Fig. 2e, top). This effect was not universal to all galectins, as binding of Gal-1 was unaffected or trended in the opposite direction (Fig. 2d, e, bottom). Thus, these data suggest that N-glycan I-branching, catalyzed by GCNT2, serves as a physiological inhibitor of Gal-9 binding to GC B cells.

**Naive B cells express Gal-9 in blood and lymphoid tissue.** The concerted upregulation of GCNT2 in GC B cells and consequent loss of Gal-9 binding suggested that Gal-9 may have important functional roles in naive and memory B cells that are switched off in GCs. To contextualize the roles of Gal-9 in vivo, we characterized Gal-9 expression in human lymphoid tissue. Immunohistochemical staining for Gal-9 protein in human tonsil, lymph node, and spleen revealed localization of Gal-9 to B cell follicles (Fig. 3a, Supplementary Fig. 5a, b). Gal-9 staining was particularly apparent in tonsil follicular mantle and was somewhat reduced in

GCs, especially in GC B cell-dense dark zones (Fig. 3a). Interestingly, Gal-9 staining was also apparent in reticular cells of the light zone and sporadically throughout the dark zone (Fig. 3a), as well cells in the medullary cords of lymph nodes (Supplementary Fig. 5a).

Although B cells had not previously been described to produce Gal-9, localization of Gal-9 to lymphoid follicles suggested that B cells themselves may serve as a significant source of Gal-9. Indeed, analysis of publicly available datasets (GSE24759)[26] revealed that peripheral blood naive B cells expressed the highest mean LGALS9 (Gal-9) transcript levels of all analyzed subsets (Supplementary Fig. 5c). Moreover, comparison of LGALS9 transcript (Fig. 3b) and Gal-9 protein levels (Fig. 3c, d) in naive, GC, and memory B cells demonstrated that Gal-9 levels were highest in naive B cells, expressed as medium- and full-length isoforms, and reduced in GC and memory B cells. Consistent with expression of linear poly-LacNAc binding determinants for Gal-9, naive and memory B cells, but not GC B cells, exhibited measurable levels of endogenous Gal-9 on their cell surface (Fig. 3d). Detection of Gal-9 on the surface of memory B cells, despite significant downregulation of Gal-9 protein, suggested that Gal-9 produced in trans may be a relevant source of Gal-9 for B cells in vivo, in addition to autologous Gal-9.

**CD45 is a major receptor for Gal-9 on naive B cells.** To gain insight into the functional importance of Gal-9, we sought to identify its major glycoprotein receptor(s) on B cells. To this end, we utilized an unbiased immunoprecipitation (IP) approach in which cell surface glycoproteins were chemically labeled with biotin prior to IP with Gal-9, followed by detection of immunoprecipitated Gal-9 receptors by western blot with streptavidin (Supplementary Fig. 5d). Using this approach, a single specific band was detectable in Gal-9 immunoprecipitates at approximately 260 kDa that was absent from control IP conditions (Supplementary Fig. 5d, red arrow). From the band's apparent molecular weight (~260 kDa), we hypothesized the band was CD45, which is expressed as a full-length isoform in B cells (CD45RABC or CD45R), is known to be heavily glycosylated, and has previously been reported to bind Gal-3 in B cell lymphoma cell lines[27]. Co-IP experiments revealed association of both recombinant (Fig. 3e) and endogenous (Supplementary Fig. 5e) Gal-9 with CD45. Additionally, treatment of naive B cells with Gal-9 resulted in visible capping of CD45 that was blocked with the inclusion of lactose (Fig. 3f). Moreover, differences in CD45 expression between naive, GC, and memory B cells failed to explain loss of Gal-9 binding to GC B cells, suggesting altered glycosylation of CD45 (via I-branching) is the principal mechanism regulating Gal-9 binding (Supplementary Fig. 5f). Collectively, these data indicate that CD45 is a major glycoprotein receptor for Gal-9 on naive B cells.

**Fig. 1** The naive to GC B cell transition is characterized by remodeling of poly-$N$-acetyllactosaminyl N-glycans (poly-LacNAc) from linear to I-branched. **a** Schematic of procedure for N-glycome analysis by MS. Human tonsil naive, GC, and memory B cells were sorted by flow cytometry before treatment with PNGase F to enzymatically release N-glycans, which were subsequently processed for MALDI-TOF MS. **b** Gating strategy for flow cytometric sorting and analysis of B cell subsets. **c** Partial MALDI-TOF mass spectra of N-glycans from B cells sorted as in **b**. Note the presence of repeating $N$-acetylglucosamine/galactose (LacNAc) units on multiple N-glycan antennae, and the additional LacNAc I-branches found on high mass N-glycans of GC B cells (compare red peaks). Structures outside a bracket have not been unequivocally defined. "M" and "m" designations indicate major and minor abundances, respectively. Full methods and complete spectra can be found in Supplementary Information. **d** Summary of proposed N-glycan structures present on naive, GC, and memory B cells. Naive and memory B cells are depicted together due to similar N-glycomic features. I-branches are shaded red. A "$+/-$" signifies that the structure can be found with or without the indicated modification. Numbers indicate number of possible LacNAc units. **e** Relative quantification of linear vs. I-branched poly-LacNAcs in naive, GC, and memory B cells. Data were computed from common ions found on all B cells and detected at $m/z$ 4402, 4675, 4763, 5124, 5212, 5573. For naive and GC B cells, data in **c-e** depict one of two experiments, each from a distinct tonsil specimen, with similar results. Data from memory B cells are from a single tonsil specimen from a single experiment

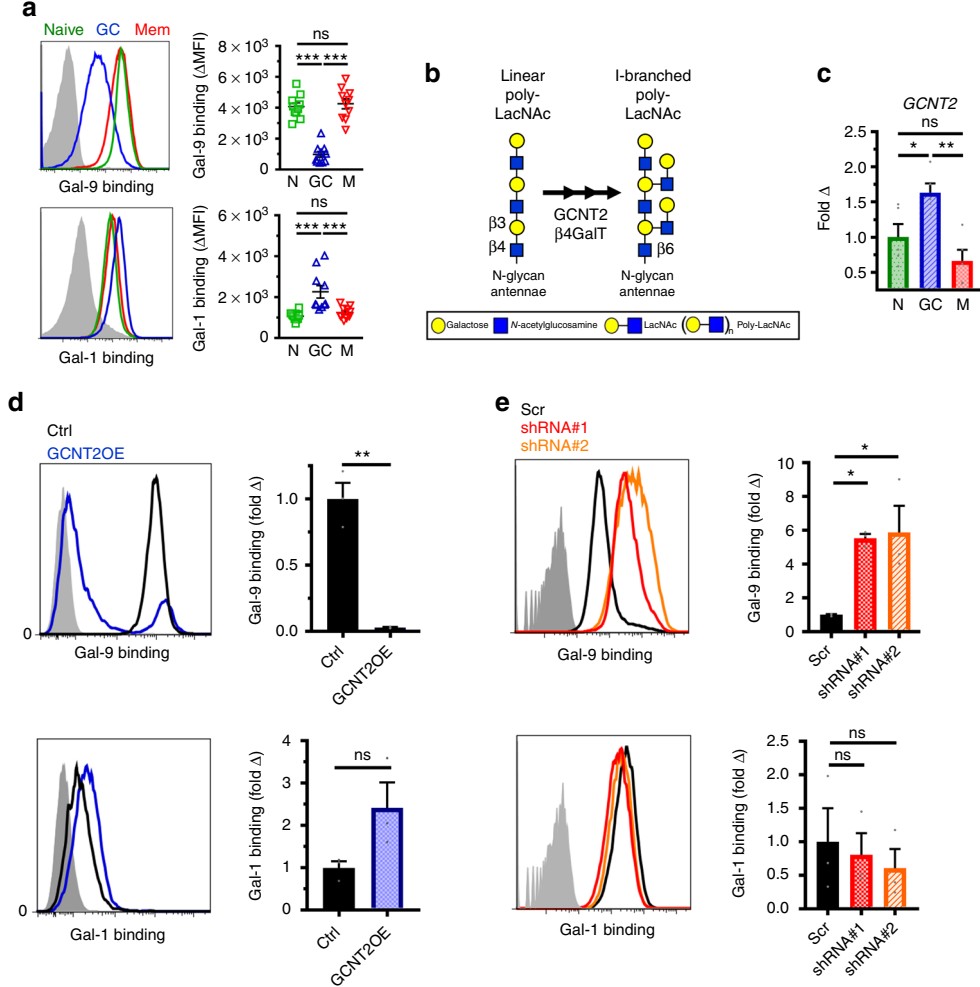

**Fig. 2** The immunomodulatory lectin Gal-9 strongly binds naive and memory B cells but is inhibited in GC B cells by I-branching of N-glycans. **a** Representative histograms (left) and quantification (right) of recombinant Gal-9 (top) and Gal-1 (bottom) by flow cytometry to tonsillar naive, GC, and memory B cells ex vivo. Gray histogram represents staining in the presence of 100 mM lactose, a competitive inhibitor of galectin binding. **b** Schematic of reported I-branch activity of the glycosyltransferase GCNT2 on N-glycans. GCNT2 initiates I-branching via transfer of an *N*-acetylglucosamine in a β1,6 linkage, followed by subsequent galactosylation by β1,4 galactosyltransferases (β4GalTs). **c** Quantitative real-time reverse-transcription PCR (qRT-PCR) analysis of *GCNT2* gene expression in human B cell subsets sorted as in Fig. 1b and Supplementary Fig. 1. **d** Representative histograms (left) and quantification (right) of recombinant Gal-9 (top) and Gal-1 (bottom) binding by flow cytometry to control (*GFP*-transduced) or *GCNT2*-transduced overexpression variant NUDUL-1 B cells. Gray histograms represent Gal-1 or Gal-9 staining in the presence of 100 mM lactose. **e** Representative histograms (left) and quantification (right) of recombinant Gal-9 (top) and Gal-1 (bottom) binding by flow cytometry to Ramos B cells transduced with control shRNA (Scr) or shRNAs targeting *GCNT2*. For **a**, $n = 10$, where each data point represents an individual tonsil specimen, pooled from two independent experiments. For **c**, $n = 5$ distinct tonsil specimens pooled from at least three independent experiments. For **d** and **e**, $n = 3$ biological replicates from three independent experiments. For **a**, **c**, and **d** statistics were calculated using one-way ANOVA with correction for multiple comparisons. For **e**, statistics were calculated using an unpaired, two-tailed *t*-test. Throughout, bars and error bars depict mean and standard error of the mean (SEM), respectively. ns = not significant, *$p \leq 0.05$, **$p \leq 0.01$, ***$p \leq 0.001$

**Gal-9 induces inhibitory signaling via Lyn, CD22, and SHP-1.** CD45 is known to regulate antigen receptor signaling through the activation of Src family kinases[28]. We therefore hypothesized that Gal-9 binding to CD45 may modulate its signal transduction activity.

To test this hypothesis, we magnetically enriched naive B cells from tonsil by negative selection, treated the cells in the presence or absence of IgM stimulation, and analyzed tyrosine phosphorylation by western blot with 4G10 mAb. We observed that Gal-9 induced dose-dependent tyrosine phosphorylation of several proteins, including a band of ~53 kDa and a prominent band of ~145 kDa (Supplementary Fig. 6a, b). These bands were consistent with the apparent molecular weights of the Src family

kinase Lyn and the inhibitory co-receptor CD22 (Siglec 2), respectively. Though Lyn is capable of initiating BCR signaling, it has been shown to favor activation of inhibitory signaling pathways through phosphorylation of immunoreceptor-based tyrosine inhibitory motifs (ITIMs) in the cytoplasmic tail of CD22[29–33], which subsequently recruits the tyrosine phosphatase SHP-1[34]. Thus, we postulated that Gal-9 binding to CD45 may induce activation of Lyn, CD22, and SHP-1. Indeed, treatment of naive B cells with Gal-9 induced dose-dependent phosphorylation of all three molecules (Fig. 4a, b) that was inhibited by inclusion of lactose. Therefore, these results suggest that Gal-9 binding to CD45 activates inhibitory signaling through the Lyn-CD22-SHP-1 pathway.

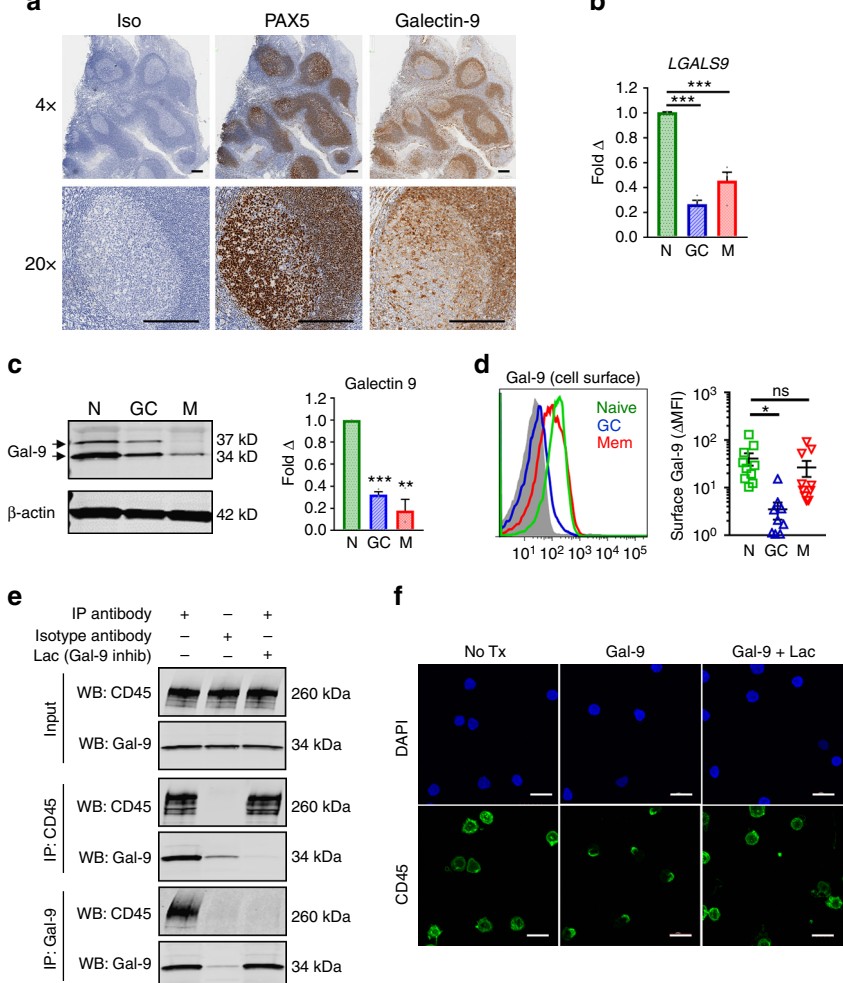

**Fig. 3** Gal-9 is expressed by naive B cells and binds the glycoprotein CD45. **a** In situ expression of Gal-9 in human tonsil. Formalin-fixed, paraffin-embedded tonsil sections were stained with Gal-9 antibody, PAX5 antibody (to identify B cells), or isotype control, followed by counterstaining with hematoxylin. Scale bar, 200 μm. **b** Relative comparison of Gal-9 expression by qRT-PCR of FACS-sorted naive, GC, memory B cells (as in Fig. 1b and Supplementary Fig. 1). Data were normalized to a housekeeping control gene *VCP* and are shown relative to naive B cells. **c** Representative western blot (left) quantification (right) of Gal-9 protein expression in lysates of FACS-sorted B cell subsets. Data were normalized to β-actin to control for differences in loading. For Gal-9 blot, the doublet band indicates expression of both medium-length and full-length Gal-9 isoforms. **d** Representative histogram (left) and quantification (right) of endogenous cell surface Gal-9 detected by ex vivo flow cytometry of unpermeabilized naive, GC, and memory B cells. **e** Co-immunoprecipitation of Gal-9 and CD45 from naive B cell lysates. Untouched naive B cells were magnetically enriched, labeled with Gal-9 (2 μg mL$^{-1}$), lysed, and immunoprecipitated/blotted with the indicated antibodies. **f** Representative immunofluorescence images of CD45 capping following Gal-9 treatment (4 μg mL$^{-1}$). Scale bar, 10 μm. For **a**, data are representative of similar results from $n = 3$ distinct tonsil specimens. For **b**, $n = 4$ separate tonsil specimens pooled from more than three independent experiments. For **c**, $n = 3$ distinct tonsil specimens pooled from three independent experiments. For **d**, $n = 10$, where each data point represents a disparate tonsil specimen pooled from two independent experiments. For **e** and **f**, data are representative of three independent experiments using three distinct tonsil specimens. For **b** and **d**, statistics were calculated using one-way ANOVA with correction for multiple comparisons. For **c**, statistics were calculated using a two-tailed, one sample *t*-test against a hypothetical value of 1. Throughout, bars and error bars depict mean and SEM, respectively. ns = not significant, $^*p \le 0.05$, $^{**}p \le 0.01$, $^{***}p \le 0.001$

**Gal-9 suppresses BCR-induced calcium signaling.** Gal-9 induction of Lyn, CD22, and SHP-1 phosphorylation suggested that Gal-9 may be a suppressor of BCR signaling. To test this, we treated magnetically enriched naive B cells with Gal-9 in the presence or absence of IgM crosslinking. Surprisingly, while Gal-9 failed to inhibit phosphorylation of several BCR signaling mediators, including CD79a, Syk, BLNK, and PLCγII, Gal-9 nevertheless strongly suppressed both nuclear factor kappa-B (NF-κB) p65 and Jun amino-terminal kinase (JNK) phosphorylation (Supplementary Fig. 6c, d). Interestingly, although Gal-9 also tended to inhibit Erk and Akt signaling in the presence of IgM stimulus, Gal-9 administered alone had the opposite effect and induced significant levels of Akt phosphorylation, while also

tending to induce phosphorylation of BLNK and Syk (Supplementary Fig. 6c, d).

NF-κB and JNK are reported to respond to fluctuations in cytoplasmic calcium through calcium-sensitive kinases; moreover, CD22 and SHP-1 signaling is known to inhibit calcium signaling through the BCR[35,36]. We therefore sought to examine whether Gal-9 may specifically influence BCR-induced calcium flux. To this end, we stimulated tonsillar B cells in the presence or absence of Gal-9 and measured accumulation of intracellular calcium by flow cytometry. Compared to crosslinking with anti-IgM F(ab')$_2$, anti-IgD F(ab')$_2$, or anti-light chain F(ab')$_2$ alone, BCR crosslinking in the presence of Gal-9 resulted in markedly suppressed peak and total calcium flux that was lactose-reversible

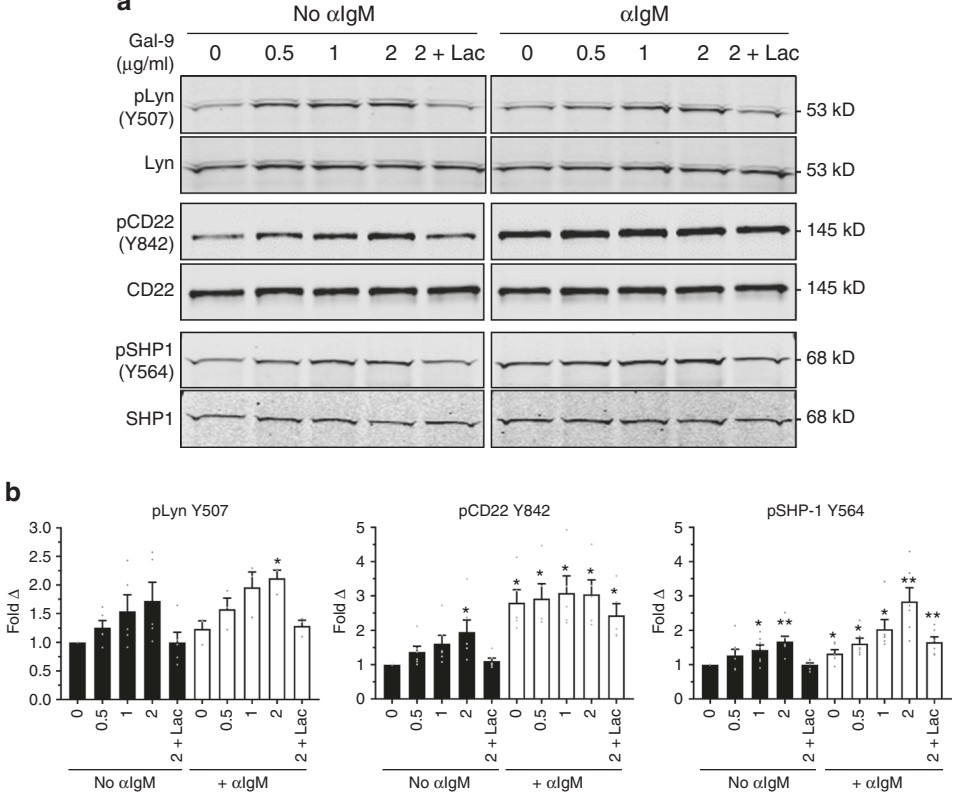

**Fig. 4** Gal-9 induces phosphorylation of Lyn, CD22, and SHP-1 in naive B cells. **a** Representative western blot and **b** quantification of Lyn, CD22, or SHP-1 phosphorylation at the indicated sites in magnetically enriched naive B cells treated with recombinant Gal-9 at the indicated concentrations in the absence (left blot) or presence (right blot) of anti-IgM F(ab')$_2$-mediated crosslinking (15 µg mL$^{-1}$) for 5 min. Blots were subsequently stripped and reprobed with antibody against total protein. Data in **b** were normalized to respective total protein control and are presented relative to no stimulation control. In all experiments, no IgM and IgM-stimulation lanes were loaded on the same gel and probed on the same blot. For **a**, data are representative of results from four independent experiments. For **b**, $n = 3$ or more distinct tonsil specimens pooled from three or more independent experiments. Statistics were calculated using a one-sample $t$-test against a hypothetical value of 1. Throughout, bars and error bars depict mean and SEM, respectively. ns = not significant, *$p \leq 0.05$, **$p \leq 0.01$, ***$p \leq 0.001$

(Fig. 5a–c; Supplementary Fig. 7b). Importantly, Gal-9 did not interfere with the binding of BCR crosslinking F(ab')$_2$ (Supplementary Fig. 7a). Although we observed that Gal-9 could induce Akt phosphorylation in the absence of BCR crosslinking (Supplementary Fig. 6d), Gal-9 by itself failed to induce significant calcium flux (Supplementary Fig. 7c).

Besides NFκB and JNK, the nuclear factor of activated T cells (NFAT) family of transcription factors are highly sensitive to fluctuations in cytoplasmic calcium levels due to calcineurin-dependent regulation of nuclear translocation[35,36]. We thus tested whether Gal-9 could prevent BCR-induced NFAT1 translocation to the nucleus in naive B cells. Strikingly, while IgM crosslinking resulted in robust NFAT1 nuclear translocation, cells co-treated with Gal-9 exhibited significantly less NFAT1 in the nucleus, as measured by both western blot of fractionated lysates (Fig. 5d, e) and immunofluorescence microscopy (Fig. 5f, g).

Taken together, these results suggest that Gal-9 inhibits BCR-mediated calcium accumulation and the subsequent activity of calcium-sensitive transcription factors.

**Removal of sialic acids ablates Gal-9 inhibitory activity**. CD45 and CD22 have been postulated to associate in resting B cells via sialic acid-dependent interactions between CD22 and the heavily sialylated ectodomain of CD45[37–39]. Our finding that Gal-9 binds CD45 and induces phosphorylation of CD22 suggested that Gal-9 inhibitory activity may depend on CD22/CD45 association. We

therefore hypothesized the suppressive activity of Gal-9 may be short-circuited by decoupling of CD22 and CD45.

To test this, we pre-treated tonsillar B cells with *Arthrobacter ureafaciens* sialidase to enzymatically remove cell surface sialic acids and measured the ability of Gal-9 to inhibit BCR calcium flux. As expected, sialidase treatment resulted in near-total loss of cell surface α2,3- and α2,6-linked sialic acids (Supplementary Fig. 7d), and removal of sialic acids did not impair Gal-9 binding to B cells (Supplementary Fig. 7e). Also as expected from previous studies, sialidase-treatment resulted in much lower (but still measurable) calcium flux, in line with reports that CD22 activity is normally restrained by *cis* associations[6,38–41] (Fig. 5h, left). Remarkably, however, while Gal-9 treatment effectively suppressed BCR calcium flux in untreated B cells, B cells pre-treated with sialidase were rendered completely insensitive to the influence of Gal-9 (Fig. 5h, middle and right). Thus, these data support a CD22-dependent mechanism of action of Gal-9 on BCR signaling.

**Gal-9 inhibits naive B cell activation**. Inhibition of BCR calcium signaling and NFAT1 nuclear translocation suggested Gal-9 may functionally suppress downstream B cell activation. To test this, we stimulated naive B cells with either T-independent stimulus (anti-IgM F(ab')$_2$ + unmethylated CpG nucleotides) or T-dependent stimulus (anti-IgM F(ab')$_2$ + recombinant CD40L-trimer (a generous gift from Dr. Gordon Freeman, DFCI[42])) in the presence or absence of Gal-9. Strikingly, inclusion of Gal-9

strongly suppressed the number of proliferating (Ki-67[+]) cells apparent at 40 h in a dose-dependent manner in both stimulation conditions (Fig. 6a, b), while having a negligible effect on B cell viability (Supplementary Fig. 7f). Gal-9 treated B cells also showed reduced blasting by FSC (Fig. 6a) and significantly lower expression of the activation marker CD86 (B7.2) (Fig. 6c, d). Gal-9 equally suppressed activation of B cells stimulated by IgM, IgD, or light chain crosslinking, suggesting that the mechanism of action in human B cells is not specific to IgM (Supplementary

Fig. 7g). In all cases, inclusion of lactose in the cell culture media, but not an inactive isomer (sucrose), was sufficient to partially or completely reverse the effects of Gal-9 (Fig. 6a–d, Supplementary Fig. 7g). Taken together, these data suggest that Gal-9 mitigates B cell activation.

**Gal-9 does not induce internalization of CD22, CD45, or BCR.** Galectins are commonly reported to regulate glycoprotein

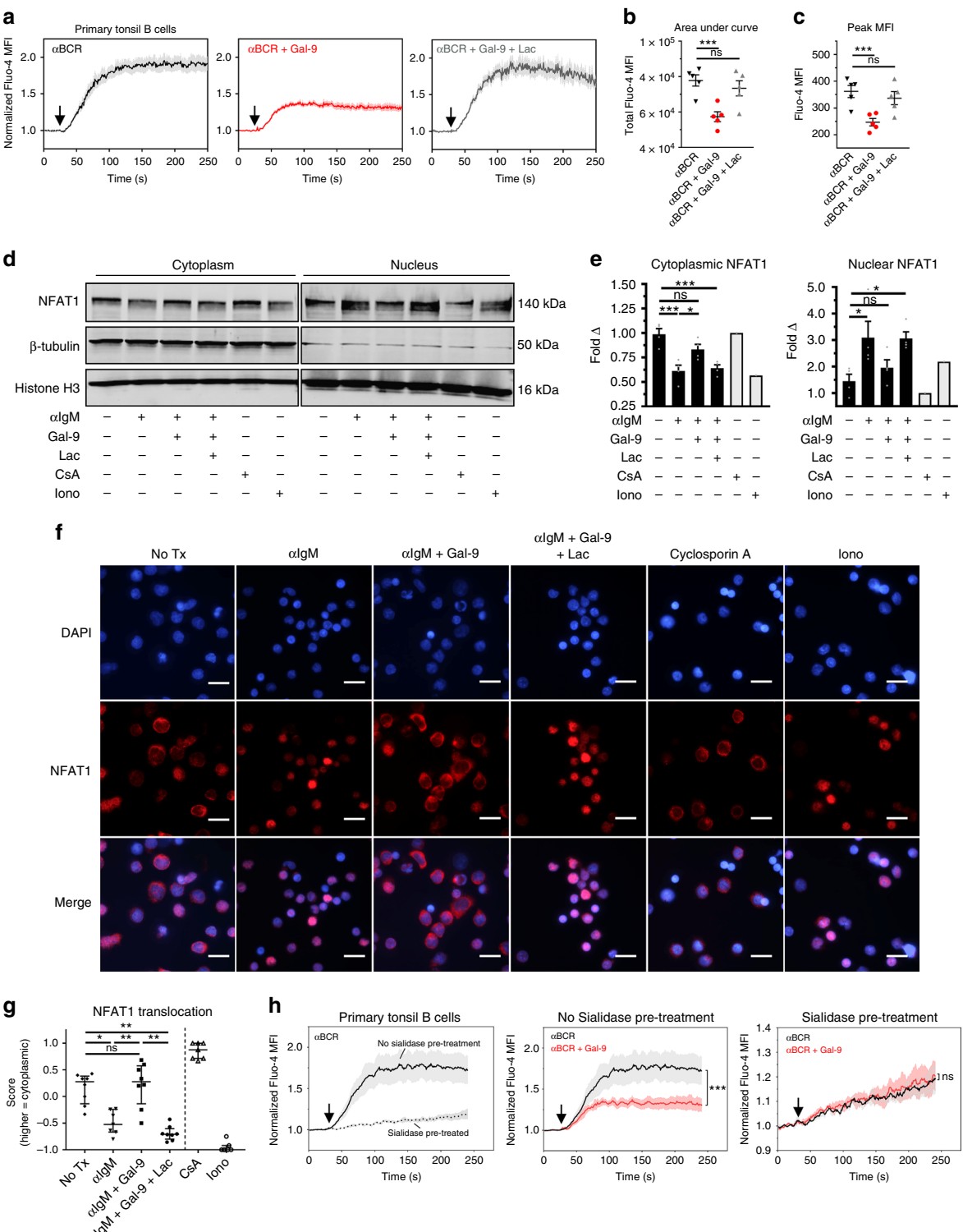

internalization. We therefore queried whether increased internalization of CD45 or CD22 could explain reduced calcium signaling associated with Gal-9 treatment. As controls, we analyzed the internalization of several other glycoproteins, including IgM, IgD, and CD19. Although we observed significant reductions in mAb binding to CD45 and CD22 after Gal-9 treatment, we found that this effect was reversed when Gal-9 was eluted with lactose just prior to Ab staining, indicating competition for Ab binding rather than bona fide internalization (Supplementary Fig. 8a). As expected, Gal-9 also did not induce internalization of IgM, IgD, or CD19 (Supplementary Fig. 8a). Therefore, these data suggest that internalization of CD45 or related glycoproteins is not responsible for Gal-9's effects on calcium signaling and activation.

**B cell-intrinsic Gal-9 autologously regulates BCR signaling**. Our findings with Gal-9-treated B cells indicated that Gal-9 is capable of suppressing BCR calcium signaling in *trans*. However, it was still unclear whether B cell-intrinsic Gal-9 could mediate similar inhibitory activity. Therefore, we tested whether elution of endogenous cell-surface Gal-9 was sufficient to augment BCR calcium flux. In support of a similar role for B cell-derived Gal-9, pre-treatment of B cells with lactose resulted in significantly higher calcium flux in response to BCR triggering (Fig. 7a).

Lactose is known to be a broad-spectrum inhibitor of galectins[5]. It was therefore possible that the improved calcium flux in lactose-treated B cells arose from indiscriminate elution of other galectin family members. To more precisely assess the effect of endogenous Gal-9 on BCR calcium flux, we generated stable Gal-9 knockdown B cell lines via lentiviral transduction with Gal-9-specific shRNAs. Initial screens of Gal-9, CD22, and SHP-1 expression in numerous B cell lines revealed significantly lower expression of CD22 and/or SHP-1 compared to primary cells in virtually all cell lines tested, and variable expression of Gal-9 that roughly corresponded with presumed cellular origin from either non-GC B cells (Gal-9hi/Gal-9mid, Karpas 1718, REC-1, SUDHL-9, NUDUL-1) or GC B cells (Gal-9lo, Ramos, Namalwa, Raji) (Fig. 7b). Because our data suggested a CD22- and SHP-1-dependent mechanism of action of Gal-9, we selected Karpas 1718 B cells for Gal-9 knockdown, due to co-expression of Gal-9, CD22, and SHP-1 (Fig. 7b, Supplementary Fig. 8b). Additionally, taking advantage of natural heterogeneity across B cell lines (Fig. 7b), we generated in parallel Gal-9 knockdown NUDUL-1 B cells, which expressed Gal-9 but lacked meaningful expression of CD22 and SHP-1, and which we reasoned would be a suitable negative control for Gal-9 activity (Fig. 7b, Supplementary Fig. 8c). Strikingly, Gal-9 knockdown Karpas 1718 B cells, but not NUDUL-1 B cells, exhibited substantially elevated calcium flux in response to BCR crosslinking, supporting an inhibitory

role for B cell-intrinsic Gal-9 that is dependent on CD22 and SHP-1 (Fig. 7c–j). Moreover, Gal-9 knockdown Karpas 1718 B cells tended to have (but not reaching significance) elevated baseline calcium levels, particularly those cells exhibiting the highest degree of knockdown (Fig. 7d, Supplementary Fig. 8b). Critically, the augmented BCR calcium response observed in Karpas 1718 Gal9 knockdown B cells was reversible upon addition of exogenous Gal-9 (Fig. 7k–m). Taken together, these data suggest that Gal-9 is both an intrinsic and extrinsic regulator of B cell activation that inhibits BCR signaling via a CD22-dependent mechanism (Fig. 8).

## Discussion

Here, we utilized whole-glycome MS to characterize the N-glycome of human naive, GC, and memory B cells. These studies unveiled a previously unappreciated role for I-branched N-glycans as a physiological regulator of binding of Gal-9, which we found was predominantly expressed by naive B cells and could potently suppress BCR calcium flux in both a B cell-intrinsic and extrinsic manner. We found that Gal-9 bound CD45, induced Lyn-CD22-SHP-1 inhibitory signaling, tempered intracellular calcium levels, attenuated NFAT1 nuclear translocation, and inhibited B cell activation.

I-branched N-glycans are among a growing list of glycan-specific features that specifically mark GC B cells, including peanut lectin (PNA) ligands (asialylated Core 1 O-glycans)[43], loss of GlcNAc sulfation[44], CD77 antigen (Gb3 glycolipid)[45], and GL7 antigen (α2,6-sialylated LacNAc)[46]. However, the specific functions of many of these glycan moieties have remained enigmatic. Our finding that I-branches regulate Gal-9 binding attributes a physiological function to a long-known developmental blood group antigen ("adult I" blood group)[24,47–51]. The observed selectivity of I-branching for Gal-9 binding, but not Gal-1 binding, is likely attributable to their respective binding modes: while Gal-9 reportedly favors binding to internal LacNAc residues of a poly-LacNAc chain, Gal-1 preferentially binds to poly-LacNAc termini[52–54].

Immunomodulatory roles for Gal-9 have previously been reported on a constellation of immune cell types, including NK cells, macrophages, eosinophils, mast cells, and T cells[16,22]. In adaptive immunity, Gal-9 has been shown to inhibit the survival and function of inflammatory T cell subsets, including Th1 T cells[55–57], Th17 T cells[58], and CD8 T cells[20,59–61], while concomitantly favoring the differentiation, stability, and effector function of regulatory T cells[58,60,62,63]. On B cells, however, though Gal-9 had anecdotally been observed to suppress humoral immune responses[17–20], a direct function and mechanism of action had heretofore not been explored. Our study demonstrates

---

**Fig. 5** Gal-9 suppresses BCR-mediated calcium signaling and nuclear translocation of NFAT1. **a** Flow cytometric measurement of B cell cytoplasmic calcium levels using Fluo-4AM indicator dye following anti-BCR F(ab')$_2$ treatment (20 µg mL$^{-1}$) in the presence or absence of recombinant Gal-9 (2.5 µg mL$^{-1}$) and/or lactose (25 mM). Data shown are gated on CD19$^+$ CD44$^{hi}$ B cells. Mean fluorescence intensities (MFI) were normalized to average MFI of 30 s baseline. Arrow, time of stimulus. **b** Area under the curve (AUC) and **c** peak MFI of data presented in **a**. **d** Representative western blot and **e** quantification of NFAT1 nuclear translocation in magnetically enriched naive B cells treated with anti-IgM F(ab')$_2$ (15 µg mL$^{-1}$) in the presence or absence of recombinant Gal-9 (2 µg mL$^{-1}$) before lysis, fractionation, and blotting with NFAT1 Ab. As controls, cells were incubated in lactose (25 mM), cyclosporin A (100 ng mL$^{-1}$), or ionomycin (1 µM). Blots were reprobed for β-tubulin and Histone H3 as loading and fractionation controls. **f** Representative immunocytochemistry and **g** quantification of NFAT1 nuclear translocation in naive B cells treated as in **d** and **e**. Scale bar, 10 µm. **h** Measurement of calcium flux in *Arthrobacter ureafaciens* sialidase-treated primary B cells, following the indicated stimulus. For **a** and **h**, solid line represents mean of five (**a**) or three (**h**) tonsil specimens from the same number of experiments. For **b** and **c**, $n = 5$, where each data point represents a distinct specimen pooled from five experiments. For **d**, data are representative of four experiments. For **e**, $n = 4$ specimens pooled from four experiments (CsA and ionomycin, $n = 1$). For **f**, data are representative of two experiments. For **g**, each data point represents average localization score for a single microscopic field (binary scoring of 1 = cytoplasmic or −1 = nuclear). Results are representative of two experiments. For **b**, **c**, and **e**, statistics were calculated using one-way ANOVA with correction for multiple comparisons. For **g**, a Kruskal–Wallis test was used with Dunn's correction for multiple comparisons. For **h**, statistics were calculated using a two-way repeated measures ANOVA using factors "time" and "treatment," with Bonferroni correction for multiple comparisons. Throughout, bars and error bars depict mean and SEM. ns = not significant, *$p < 0.05$, **$p < 0.01$, ***$p < 0.001$

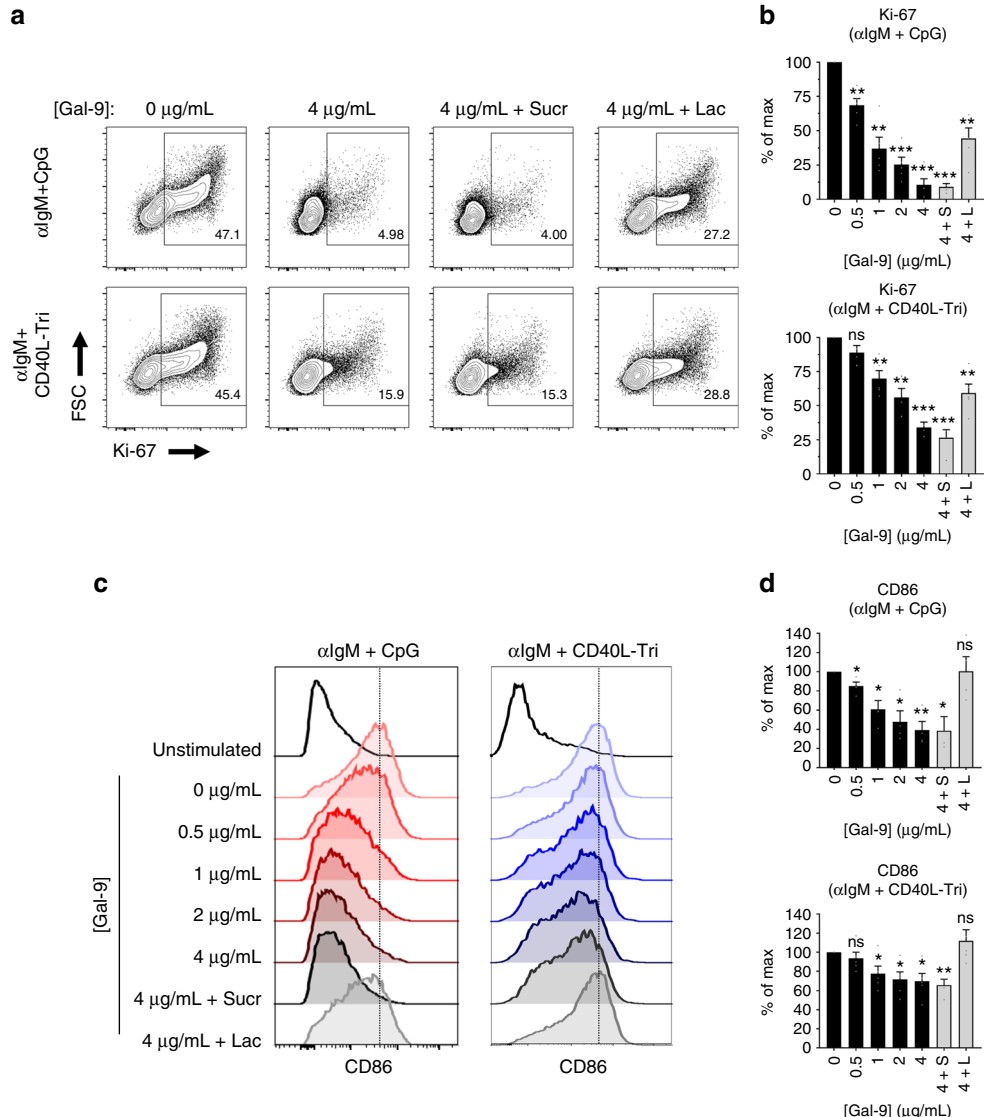

**Fig. 6** Gal-9 inhibits B cell activation and proliferation. **a** Representative contour plots and **b** quantification of naive B cell proliferation 40 h post-activation with either T-independent (anti-IgM F(ab')$_2$ + unmethylated CpG oligonucleotides + IL-2/4/10) or T-dependent stimulation (anti-IgM F(ab')$_2$ + recombinant CD40L-trimer + IL-2/4/10/21), in the presence or absence of the indicated concentration of recombinant Gal-9. "4 + L" indicates 4 μg Gal-9 plus 10 mM lactose; "4 + S" indicates 4 μg Gal-9 plus 10 mM sucrose, a non-inhibitory osmolarity control. **c** Representative histograms and **d** quantification of CD86 (B7-2) expression on naive B cells 40 h after activation in the presence or absence of Gal-9 at the indicated concentrations, as in **b**. For **a** and **c**, data are representative of five independent experiments. For **b** and **d**, $n = 5$ distinct tonsil specimens pooled from five independent experiments. For **b**, and **d**, statistics were calculated using a one-sample $t$-test against a hypothetical value of 100. Throughout, bars and error bars depict mean and SEM, respectively. ns = not significant, $*p \leq 0.05$, $**p \leq 0.01$, $***p \leq 0.001$

that Gal-9 is capable of directly suppressing BCR calcium flux and activation by binding CD45 and activating Lyn-CD22-SHP-1 inhibitory signaling.

How Gal-9 transduces a signal through CD45 to activate Lyn and/or CD22 is not clear, but likely involve one of two possibilities: (1) Ligand/receptor style activation of CD45 enzymatic activity and subsequent activation of Lyn; (2) Altered CD45 and CD22 membrane distribution to compartments containing Lyn. The first model is supported by a previous study showing that Ab crosslinking of CD45 induces CD22 phosphorylation in a remarkably similar fashion to our own findings[64]. Feasibly, Gal-9 may function as a physiological ligand for CD45 in a similar manner to other receptor-type protein tyrosine phosphatases[28]. Arguably, however, the second model best fits with reported galectin biology, particularly the ability of galectins to regulate glycoprotein membrane dynamics through redistribution into

galectin-glycoprotein networks termed "galectin lattices"[65]. Increasing evidence suggests that BCR, Lyn, CD45, and CD22 are spatially organized into discrete micro- and nano-domains at steady state[38,39,66]. In this regard, Gal-9 may promote redistribution of CD45 (and by association, CD22[38,39]) to domains containing Lyn[67], which could initiate inhibitory signaling by phosphorylating CD22 ITIMs. This model is supported by similar reports of galectin-mediated regulation of CD45 membrane compartmentalization in T cells[68]. Additional studies, in particular those assessing localization of CD45, CD22, and IgM by super-resolution microscopy, will be necessary to determine the effect of Gal-9 on the membrane distribution of these glycoproteins.

Our finding that Gal-9 specifically inhibits calcium signaling is surprising given the well-established roles for SHP-1 in targeting proximal and intermediate regulators of BCR signaling, including

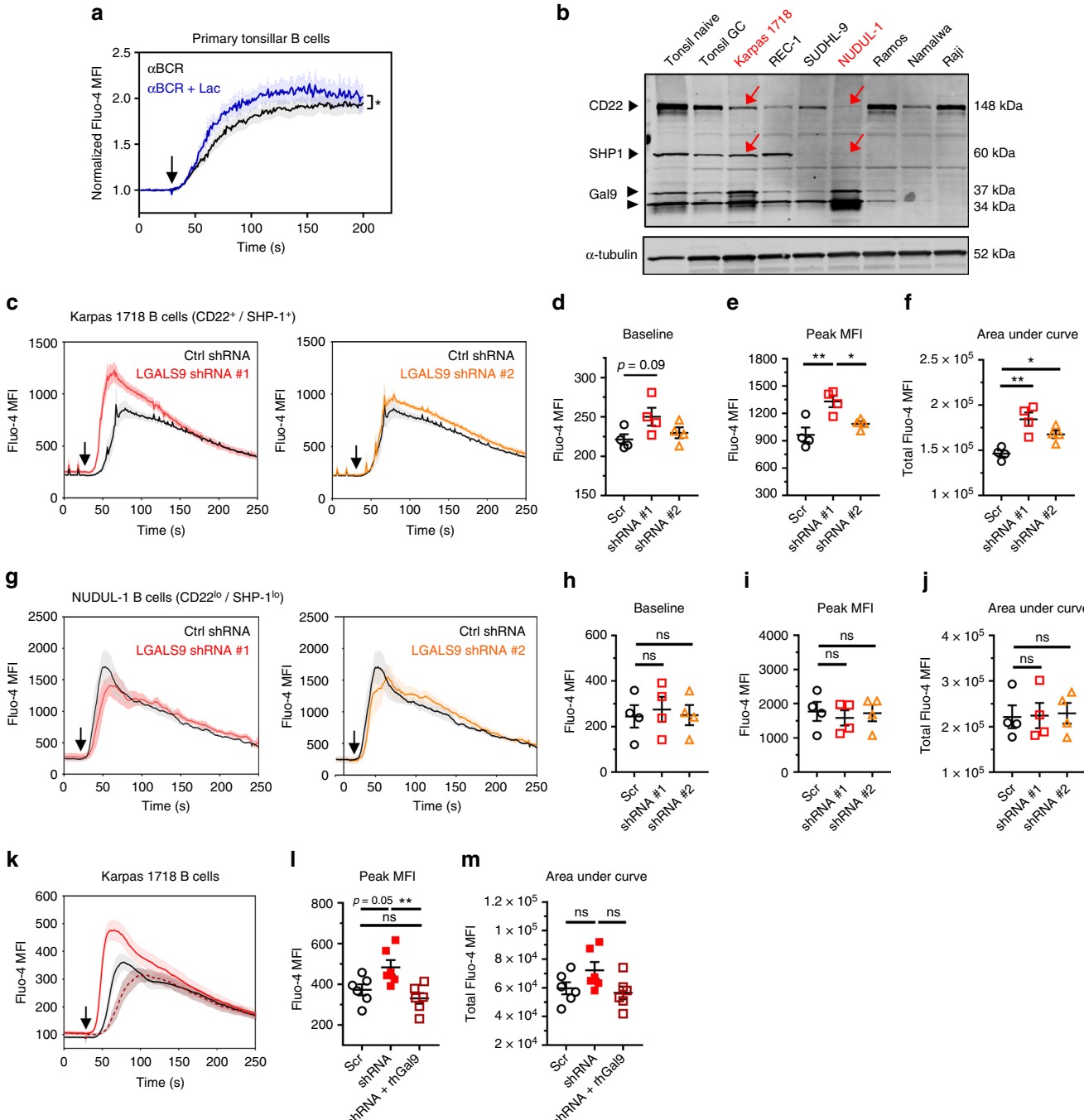

**Fig. 7** B cell-intrinsic Gal-9 is a negative regulator of BCR calcium signaling. **a** Flow cytometric assessment of B cell cytoplasmic calcium levels using Fluo-4AM indicator dye following anti-BCR F(ab')$_2$ stimulation (20 μg mL$^{-1}$) after 90 min culture in the presence or absence of 25 mM lactose. Depicted are results from CD44$^{hi}$ CD19$^+$ B cells. Mean fluorescence intensities (MFI) over the entire acquisition period were normalized to the average MFI of the 30 s baseline. Arrow, time of stimulus. **b** Western blot of lysates from indicated primary cells or B cell lines with the specified combination of Abs. **c** Analysis of calcium flux following anti-BCR F(ab')$_2$ stimulation (20 μg mL$^{-1}$) in Karpas 1718 B cells (CD22$^+$ and SHP-1$^+$) transduced with either control shRNA (Scr) or one of two shRNAs against *LGALS9*. **d** Pre-stimulation baseline, **e** peak, or **f** total calcium flux from experiments presented in **c**. **g** Analysis of calcium flux following anti-BCR stimulation (20 μg mL$^{-1}$) in NUDUL-1 B cells (CD22$^{lo}$ and SHP-1$^{lo}$) transduced with either control shRNA (Scr) or one of two shRNAs against *LGALS9*. **h** Pre-stimulation baseline, **i** peak, or **j** total intracellular calcium levels from experiments presented in **g**. **k** Analysis of calcium flux following anti-BCR stimulation (20 μg mL$^{-1}$) as in **c**, except that 10 μg mL$^{-1}$ recombinant Gal-9 was co-administered with BCR stimulus where indicated. **l** Peak or **m** total intracellular calcium levels from experiments presented in **k**. For **a**, **c**, **g**, and **k**, the solid line represents the mean and shaded error bars represent SEM from five (**a**), four (**c** and **g**), or six (**k**) different biological replicates pooled from the same number of independent experiments. For **d**–**f** and **h**–**j**, n = 4, where data points depict biological replicates pooled from four independent experiments. For **l**–**m**, n = 6, where data points depict biological replicates pooled from six independent experiments. For **a**, statistics were calculated using a two-way repeated measures ANOVA using factors "time" and "treatment," with Bonferroni correction for multiple comparisons. For **d**–**f**, **h**–**j**, and **l**–**m**, statistics were calculated using one-way ANOVA with correction for multiple comparisons. Throughout, bars and error bars depict mean and SEM, respectively. ns = not significant, *$p \leq 0.05$, **$p \leq 0.01$, ***$p \leq 0.001$

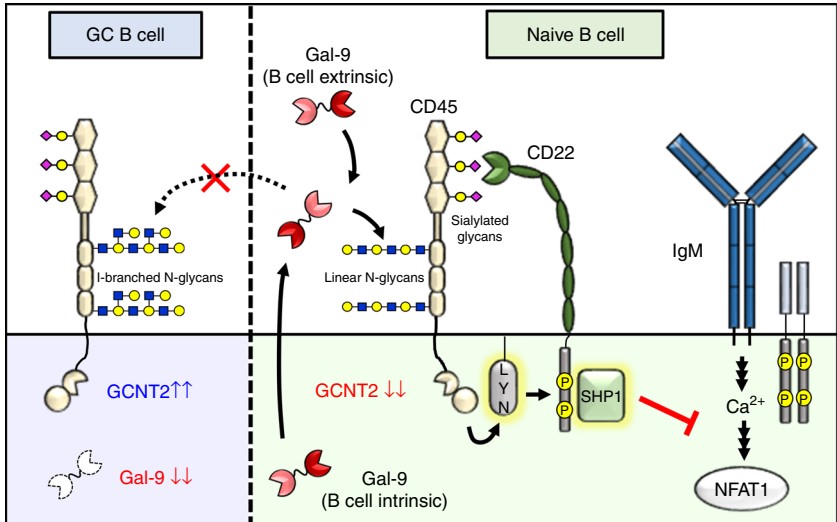

**Fig. 8** Proposed role of Gal-9 and I-branches in the regulation of BCR signaling in naive B cells vs. GC B cells. In naive B cells (right), Gal-9 is expressed at high levels and secreted into the microenvironment by an unconventional mechanism. Autologously, or exogenously, produced Gal-9 binds poly-LacNAc-containing N-glycans on CD45, which are predominantly of the linear type due to low expression of the I-branching glycosyltransferase GCNT2. Gal-9 binding to CD45 activates Lyn (by an undetermined mechanism), which subsequently phosphorylates tyrosine residues in CD22 ITIMs and recruits the protein tyrosine phosphatase SHP-1. SHP-1 phosphatase activity dampens cytoplasmic calcium levels, including calcium accumulation in response to BCR engagement, possibly through its reported ability to activate B cell calcium efflux pumps. Diminished intracellular calcium levels results in decreased nuclear translocation and activity of NFAT1 and other calcium sensitive signaling factors, ultimately inhibiting B cell activation. By contrast, Gal-9 activity is downmodulated in GC B cells (left) via the combined downregulation of Gal-9 protein and upregulation of *GCNT2*, which disfavors Gal-9 binding by modifying N-glycan poly-LacNAcs with I-branches

CD79, Syk, and BLNK[28]. Several lines of reasoning may explain our observations. First, in B cells, CD22 and SHP-1 have been described to associate with and activate the plasma membrane calcium efflux pump, isoform 4a (PMCA4a)[69]. It is possible that Gal-9 may specifically promote association of CD45 (and thereby CD22) with PMCA4a. Second, Lyn is known to be a dual regulator of BCR signaling, promoting both positive signaling by acting on ITAM-containing molecules such as CD79 and CD19, and inhibitory signaling via phosphorylation of ITIMs in CD22[33]. Plausibly, the net phosphorylation state for positive BCR signaling regulators, including BLNK, Syk, and PLCγII, may appear unchanged due to countervailing actions of Lyn and SHP-1. By contrast, inhibitory factors such as PMCA4a may be shifted toward activation. Third, because galectins have previously been shown to bind an array of glycoproteins, Gal-9 may possibly interact with other glycoprotein receptors at a level below the limit of detection in our IP discovery approach (Supplementary Fig. 5d). Finally, we cannot rule of the possibility that under the relatively strong stimulation conditions used in this study (10–20 µg mL$^{-1}$ anti-BCR F(ab')$_2$), effects of Gal-9 on non-calcium dependent BCR signaling pathways may have been inadvertently obscured.

Our finding that Gal-9 can act as both an intrinsic and extrinsic regulator of BCR signaling is consistent with the autocrine and paracrine functions described for galectins[5,16]. Because we observed that Gal-9 is capable of regulating calcium signaling in a B cell-intrinsic manner, it is tempting to speculate that Gal-9 may contribute to maintenance of B cell peripheral tolerance by modulating the threshold of activation. In this regard, Gal-9 activity may be dynamically tuned higher or lower by altering Gal-9 secretion, which occurs by an unconventional and still-undefined mechanism, or Gal-9 expression. In contrast to naive B cells, we found memory B cells downregulated Gal-9 while preserving binding to exogenous Gal-9. An interesting possibility arising from this finding is that the downregulation of Gal-9 may contribute to the lower activation threshold of memory B cells.

Moreover, Gal-9 produced by other B cells or regulatory immune cells may act to elevate the activation threshold of both naive and memory B cells and/or serve as a contraction mechanism for ongoing B cell immune responses. Additional studies will be needed to parse the relative contribution of Gal-9 presented in *trans* vs. *cis* to B cell function.

A major unresolved question and area for follow-up study is why Gal-9 and Gal-9 binding are downregulated in GC B cells. This question is particularly intriguing due to possible Gal-9 expression differences between light zones and dark zones of tonsillar GCs (Fig. 3a), and when considered with reports of similar downregulation of both CD22 ligands and SHP-1 in GCs[44,46,70]. The parallel downregulation of multiple components of the Gal-9/CD22/SHP-1 mechanism proposed here supports the idea that BCR signaling, and in particular calcium signaling, may be fundamentally altered in GC B cells[71]. Indeed, in line with this notion, anecdotal reports suggest that GC B cells exhibit greater nuclear localization of NFAT1 compared to non-GC cells[72]. Clearly, additional studies will be required to parse the roles of Gal-9 and more broadly, BCR calcium signaling, in GC biology.

Clinically, Gal-9 has been reported to be elevated in serum during inflammation, including during infection, suggesting that systemically expressed Gal-9 may promote resolution of B cell responses or perpetuate B cell dysfunction during chronic disease[22]. In addition, Gal-9 has received considerable attention as a possible therapeutic target in cancer, predominantly for its inhibitory effects on TIM-3$^+$ T cells[21,22]. Our findings suggest that Gal-9's influence on the B cell compartment will also need to be carefully considered. Our study also adds to mounting evidence that Gal-9 may be harnessed therapeutically to alleviate autoimmune disease[17,18,23], or antagonized to promote B cell activation and Ab responses. Indeed, administration of Gal-9 has been shown to mitigate disease severity in preclinical models of lupus[18,23], and Gal-9-deficient mice exhibit enhanced Ab responses to infection[20] and immunization[19]. Finally, given the

effect of GCNT2/I-branches on Gal-9 but not Gal-1 binding activity, manipulating GCNT2 expression or activity may be a viable strategy to modulate susceptibility to Gal-9 in a range of clinical settings.

In summary, using an N-glycomics approach, we have uncovered a novel BCR regulatory axis involving Gal-9 that is endogenously tuned through the action of the I-branching enzyme GCNT2. Our findings add to the growing body of literature highlighting the critical roles of glycans and glycan-binding proteins in the regulation of innate and adaptive immunity.

## Methods

**Contact for reagent and resource sharing.** Requests for reagents and additional information should be directed to corresponding authors, Charles J. Dimitroff (cdimitroff@bwh.harvard.edu) or Stuart M. Haslam (s.haslam@imperial.ac.uk).

**Antibodies and reagents.** Detailed information about antibodies and other reagents used in this study are available in Supplementary Table 2.

**Cell lines.** Ramos, Raji, and NUDUL-1 B cells were a generous gift from Dr. Shiv Pillai (Ragon Institute of MGH, MIT, and Harvard, Boston, MA). Karpas 1718 B cells were generously provided by Dr. José Martinez-Climent (University of Navarra, Pamplona, Spain). SUDHL-9 B cells were a generous gift from Dr. Alan Epstein (USC Keck School of Medicine, Los Angeles, CA). REC-1 (CRL-3004) and Namalwa B cells (CRL-1432) were purchased from ATCC. All B cell lines were maintained at $0.5 \times 10^6$ – $2.0 \times 10^6$ cells mL$^{-1}$ in complete RPMI medium (RPMI 1640 + 10% (v/v) FBS + 25 mM HEPES + 1% (v/v) Penicillin/Streptomycin). Cell lines were not independently confirmed to be mycoplasma-free by our laboratory, although no signs of apparent contamination were observed. Validation of cell lines was performed in-house by RNA and/or protein expression analysis of numerous lineage markers, including CD10, CD19, CD20, CD22, IgM, IgD, IgG, and/or Bcl-6, and results were compared to expression profiles reported by cell line repositories (ATCC and DSMZ) and in publicly available datasets (GSE42204 and GSE2350). Media was renewed every 2-3 days (Ramos, Raji, Namalwa) or every 3-5 days (REC-1, NUDUL-1, and Karpas 1718).

To generate stable Ramos, Karpas 1718, and NUDUL-1 knockdown cell variants, the following shRNA constructs in the pLKO.1-puro vector were used (all from Sigma): human *GCNT2* shRNA #1: 5′-GCTCACCTCTATATTAGTTTA-3′; human *GCNT2* shRNA #2: 5′-GCTAACAAGTTTGAGCTTAAT-3′; human *LGALS9* shRNA #1: 5′-TGGTCAGCACCTGTTTGAATA-3′; human *LGALS9* shRNA #2: 5′- CCCTCCTCTCTGACCTTTAAC-3′; Non-target shRNA ("scrambled"): 5′-CAACAAGATGAAGAGCACCAA-3′. Lentivirus containing each shRNA construct was produced by co-transfection of the helper plasmids, pMD2.G-VSV-G and psPAX2-Δ8.9 into the packaging cell line HEK293T using TransIT-LT1 transfection reagent (Mirus Bio) according to the manufacturer's protocol. For lentiviral transduction, $1 \times 10^6$ Ramos, NUDUL-1, or Karpas 1718 B cells were mixed with 1 mL lentiviral supernatant containing 8 μg mL$^{-1}$ polybrene, followed by centrifugation ("spinfection") at 1000×*g* for 1 h at room temperature. Cells were subsequently cultured in fresh media for 24 h, followed by selection and maintenance in puromycin dihydrochloride (Corning).

NUDUL-1 *GCNT2* overexpression cells were generated using similar lentiviral transduction techniques as above. *GCNT2* pLenti6.3-DEST overexpression plasmid was generated using LR clonase (Invitrogen) to recombine pLenti6.3-DEST (Invitrogen) with pDONR223-GCNT2 (hORFeome ID: 54744—based on MGC:163396), according to the manufacturer's recommendations. Lentivirus was generated in transfected HEK293T cells as described above. NUDUL-1 B cells were spinfected as above with either *GCNT2* virus or *GFP* control virus, followed by selection and maintenance of overexpression cells in blasticidin S HCl (Invitrogen).

**Tonsil processing, cryopreservation, and thawing.** Fresh, anonymized discarded tonsil specimens were obtained from patients undergoing routine tonsillectomies at Children's Hospital Boston and Faulkner Hospital, in accordance with the Partners Institutional Review Board (IRB), which deemed the research as not meeting the definition of human subjects research. Tonsils were briefly (<1 h) stored on ice in isotonic saline solution before being transferred to Hank's Balanced Salt Solution (HBSS) for processing. To isolate mononuclear cells, tonsils were minced in HBSS, mashed and strained through a 70 μm nylon mesh with a 5 mL syringe plunger, rinsed two times with HBSS, and removed to a conical tube stored on ice. The procedure was repeated for the remaining tissue until the tonsil was fully processed. The resulting cell suspension was separated by density gradient centrifugation through Histopaque 1077 (Sigma) at 1000×*g* in an Allegra X-12R centrifuge, without the brake. Mononuclear cells were removed from the interface, washed 3× with cold HBSS, and frozen in 90% FBS/10% DMSO freezing media at −80 °C. For long term storage, cells were stored in a vapor-phase of a liquid nitrogen freezer. Cryopreserved tonsil mononuclear cells were rapidly thawed in a 37 °C water bath,

and warmed, complete RPMI (RPMI 1640 + 10% FBS + 1% Penicillin/Streptomycin + 25 mM HEPES) was added dropwise. Cells were then washed and counted for subsequent use. Cell viability was checked with propidium iodine. Viability was routinely >80%.

**Flow cytometry sorting.** Tonsil mononuclear cells were thawed, washed, and counted as described above. To stain apoptotic and necrotic cells, cells were incubated with Zombie NIR fixable viability dye (Biolegend) for 15 min at room temperature. Cells were washed, resuspended in a surface stain cocktail containing anti-IgD-FITC, anti-CD38-PE, anti-CD27-PE/Cy7, anti-CD19-APC, anti-CD3-APC/Cy7, and anti-CD14-APC/Cy7 (all from Biolegend), and incubated for 45 min on ice. Subsequently, cells were washed two times, passed through a 35 μm nylon mesh and sorted on a BD FACS Aria II at the Harvard Microbiology and Immunobiology (MBIB) Flow Cytometry Core. Lymphocytes were electronically gated according to their forward scatter and side scatter properties, negatively gated to exclude viability dye+, CD3+, and CD14+ cells, and positively gated for B cells (CD19+). B cell subsets were subsequently gated as follows: IgD+CD27- cells (Naive); IgD- CD38hi CD27+/- (GC); IgD- CD38lo/mid CD27+ (Memory) (See also Fig. 1b and Supplementary Fig. 1). The CD27 gate was set based on a PE/Cy7 FMO gating control. Sorted cells were pelleted, washed 2× with PBS, then lysed in Buffer RLT (Qiagen, for RNA analysis), 2% NP-40 lysis buffer (for western blot), or snap frozen in an isopropanol/dry ice slurry (for glycomic analysis).

**Magnetic enrichment of naive B cells by negative selection.** To enrich naive B cells by negative selection, tonsil mononuclear cells were thawed, washed, counted, resuspended in MACS buffer (PBS + 0.5% BSA + 2 mM EDTA) and labeled for 10 min on ice with B cell isolation kit II antibody cocktail (Miltenyi) supplemented with the following additional antibodies: CD10-biotin (Miltenyi), CD27-biotin (Biolegend), CD77-FITC (Biolegend). Cells were then washed, labeled with anti-biotin and anti-FITC microbeads (Miltenyi), and incubated on ice for 20 min. After washing, the cells were resuspended in MACS buffer, loaded onto one or more LS columns (Miltenyi) placed in MACS separation magnets (Miltenyi), and the flow-through (unlabeled fraction) was collected. Post-sort B cell purity was confirmed on a FACS Canto I using the flow cytometry staining procedures described above, and were at least 85% pure, but typically >90%. Where indicated, to obtain a GC-depleted fraction, sorts were performed as above, except that CD27 antibody was omitted.

**N-glycome analysis by mass spectrometry.** FACS-sorted human tonsillar Naive, GC, and Memory B cells (Fig. 1b) were sorted to >95% purity (Supplementary Fig. 1) as described above. Sorted B cells were washed three times with PBS before being snap frozen in a dry ice/isopropanol slurry.

Cell pellets were treated as described previously[73,74]. Briefly, cell pellets ($4.7 \times 10^6$ cells for naive, $2.8 \times 10^7$ cells for GC, and $2.5 \times 10^7$ cells for Memory) were subjected to sonication in the presence of detergent (CHAPS), reduced in 4 M guanidine-HCl (Pierce), carboxymethylated, and digested with trypsin. The digested glycoproteins were then purified by C18-Sep-Pak (Waters Corp.). N-glycans were released by peptide *N*-glycosidase F (E.C. 3.5.1.52; Roche Applied Science) digestion. Released N-glycans were permethylated using the sodium hydroxide procedure and purified by C18-Sep-Pak. The results shown are representative of two independent cell glycan preparations.

To analyze the structure of released glycans, matrix-assisted laser desorption ionization time of flight mass spectrometry (MALDI-TOF MS) and MALDI-TOF/TOF MS/MS were performed. MS and MS/MS data were acquired using a 4800 MALDI-TOF/TOF (Applied Biosystems) mass spectrometer. Permethylated samples were dissolved in 10 μl of methanol, and 1 μl of dissolved sample was premixed with 1 μl of matrix (10 mg mL$^{-1}$ 3,4- diaminobenzophenone in 75% (v/v) aqueous MeCN), spotted onto a target plate, and dried under vacuum. For the MS/MS studies, the collision energy was set to 1 kV, and argon was used as collision gas. The 4700 Calibration standard kit, calmix (Applied Biosystems), was used as the external calibrant for the MS mode, and [Glu1] fibrinopeptide B human (Sigma) was used as an external calibrant for the MS/MS mode.

For analysis of mass spectra, data were processed using Data Explorer 4.9 Software (Applied Biosystems). The processed spectra were subjected to manual assignment and annotation with the aid of a glycobioinformatics tool, GlycoWorkBench[75]. The proposed assignments for the selected peaks were based on $^{12}$C isotopic composition together with knowledge of the biosynthetic pathways. Proposed structures were then confirmed by data obtained from MS/MS and linkage analysis experiments.

Further glycan structure interrogation was performed using gas chromatography-MS (GC-MS) linkage analysis. Partially methylated alditol acetates were prepared as previously described[73]. Linkage analysis of partially methylated alditol acetates was performed on a PerkinElmer Life Sciences Clarus 500 instrument fitted with a RTX-5 fused silica capillary column (30 m x 0.32 mm inner diameter; Restek Corp.). The sample was dissolved in 20–50 μl of hexanes and injected manually (2–3 μl). Injector temperature was set at 250 °C. Partially methylated alditol acetates were eluted with the following linear gradient oven. Initially the oven temperature was set at 65 °C for 1 min, heated to 290 °C at a rate

of 8 °C per min, held at 290 °C for 5 min, and finally heated to 300 °C at a rate of 10 °C per min.

**Plant lectin and galectin staining by flow cytometry.** Tonsil mononuclear cells were thawed, washed, and counted as described above before being stained in a series of steps for viability, lectin binding, lectin detection, and lineage markers. To assess viability, cells were first incubated with Zombie NIR fixable viability dye (Biolegend) in PBS for 15 min at room temperature. For lectin stains, cells were resuspended in recombinant, *E.coli-* derived human Gal-9 (R&D,1 μg mL$^{-1}$ in 1% BSA), recombinant, *E.coli*-derived Gal-1 (Peprotech, 50 μg mL$^{-1}$), or biotinylated plant lectin (Vector) for 45 min on ice. Afterwards, cells were washed and resuspended in anti-human Gal-9-APC (Biolegend, 2 μg mL$^{-1}$), Gal-1 antibody (R&D, 2.5 μg mL$^{-1}$), or streptavidin-APC conjugate (Biolegend) in 1% BSA/PBS for 30 min on ice. For Gal-1 stains, Gal-1 antibody was subsequently detected with donkey anti-goat APC (R&D) for 25 min on ice. After lectin staining, cells were washed and stained with a panel of lineage-specific antibodies, including IgD-FITC, CD38-PE, CD19-PerCP, CD27-PE/Cy7, CD3-APC/Cy7, and CD14-APC/Cy7 (all from Biolegend) for 45 min on ice, then washed and acquired immediately on a BD FACS Canto I. Cells were gated as in Fig. 1b and Supplementary Fig. 1. For endogenous Gal-9 surface staining, no recombinant Gal-9 was added. For negative controls of Gal-1 and Gal-9 staining, a competitive galectin inhibitor, 100 mM lactose, was included in recombinant galectin stains, anti-galectin antibody stains, and appropriate wash steps. For CD27 stains, a PE/Cy7 FMO gating control was employed. For calculation of MFIs, the geometric mean was used. For cell line galectin and lectin stains, the same protocol was used, except the antibody surface stain was omitted.

**B cell activation and proliferation assessments.** Naive B cells were magnetically enriched to >85% purity by negative selection as described above, washed, then resuspended at $2 \times 10^6$ cells mL$^{-1}$ in complete Iscove's Modified Dulbecco's Medium (10% FBS, 1% Penicillin/Streptomycin, 15 μg mL$^{-1}$ gentamycin, 25 mM HEPES). Cells were subsequently treated for 40 h with varying concentrations of Gal-9 plus one of two stimulus cocktails: a T-dependent activation cocktail (Goat anti-human IgM F(ab')$_2$ (10 μg mL$^{-1}$) + CD40L recombinant trimer[42] (1 μg mL$^{-1}$) + IL-2 (10 ng mL$^{-1}$) + IL-4 (20 ng mL$^{-1}$) + IL-10 (20 ng mL$^{-1}$) + IL-21 (50 ng mL$^{-1}$)) or T-independent activation cocktail (Goat anti-human IgM F(ab')$_2$ (10 μg mL$^{-1}$) + ODN2006 (100 nM) + IL-2 (10 ng mL$^{-1}$) + IL-4 (20 ng mL$^{-1}$) + IL-10 (20 ng mL$^{-1}$)). As a negative control for Gal-9 activity, cells were treated in parallel in the presence of the competitive inhibitor lactose (10 mM) or osmolarity control sucrose (10 mM) where indicated. In some experiments, where indicated, anti-IgD F(ab')$_2$ or anti-κ/λ light chain F(ab')$_2$ (both Southern Biotech) were used rather than anti-IgM F(ab')$_2$. Following the 40 hr activation period, cells were pelleted, washed, stained with Zombie NIR viability dye, then stained with CD86 antibody (Biolegend) for 45 min on ice in FACS buffer (2% (v/v) FBS in PBS). Simultaneously, a portion of the activated B cells were permeabilized for 30 min on ice using the FoxP3/Transcription Factor Fixation/Permeabilization Kit (eBioscience), per the manufacturer's instructions. Permeabilized cells were subsequently stained with Ki-67 antibody (Biolegend) for 45 min on ice. All cells were washed two times and acquired immediately on a FACS Canto I. Cytokines were all purchased from Biolegend. CD40L-Tri was a generous gift from Dr. Gordon Freeman, Dana Farber Cancer Center (Boston MA)[42]. ODN2006 was purchased from Invivogen.

**Quantitative real-time reverse-transcription PCR (qRT-PCR).** B cell subsets were sorted to >95% purity by flow cytometry as described above, or washed and pelleted from cell culture (B cell lines), and total RNA was isolated using the RNeasy Mini RNA isolation kit (Qiagen) according to the manufacturer's instructions. RNA concentration and purity was checked using a BioDrop μLITE, and 0.25 μg RNA per reaction was subsequently converted to cDNA using the SuperScript VILO cDNA synthesis kit (ThermoFisher), per the manufacturer's instructions. Samples were assayed using Fast SYBR Green Master Mix (Applied Biosystems), and kinetic PCR was performed on a StepOne Plus Real-Time PCR System (Applied Biosystems). Samples were assayed in triplicate. Data were normalized to the housekeeping gene *VCP*. Relative transcript levels were analyzed using the $2^{(-\Delta\Delta Ct)}$ method[76]. Primer sequences used can be found in Supplementary Table 1.

**Western blot.** For analysis of total protein expression, B cells were sorted by flow cytometry as described above, washed, and lysed for 1 h on ice in 2% NP-40 buffer/ Buffer A (150 mM NaCl, 0.5 mM Tris, 1 mM EDTA) supplemented with protease/ phosphatase inhibitors (Protease/ Phosphatase Inhibitor Mini tablets, Thermo). Alternatively, various B cell lines were pelleted, washed 2×, and lysed as above. Lysates were cleared of debris by centrifugation, and either stored at −80 °C for future analysis or used immediately. Lysates were boiled for 10 min in Laemlli reducing sample buffer, and an equal amount of lysate per lane (determined by BCA assay (Thermo)) was loaded onto 4–20% Criterion Tris-HCl polyacrylamide gels (BioRad). Gels were transferred to nitrocellulose membranes (0.2 μm pore size) for western blotting. Membranes were blocked in Odyssey Blocking buffer (Li-cor) for 1 h, blotted overnight (or as described in Supplementary Table 2) with the

indicated primary antibody, and detected with species-specific IRDye 800CW or 680RD antibody conjugates (Li-Cor) for a half hour. Antibodies were diluted in 1:1 Odyssey Blocking buffer/TBS-Tween 20 (0.1%). Blots were scanned using a Li-Cor Odyssey CLx Near-infrared Imaging System. After scanning, blots were reprobed with antibodies against protein loading control (β-actin or α-tubulin). Band intensities were calculated using ImageStudio software (Li-Cor), and signal was normalized to housekeeping protein levels in each lane. Uncropped western blots are available in Supplementary Figure 9.

**Immunohistochemistry.** Formalin-fixed, paraffin embedded tonsil tissue was prepared from fresh tonsil tissue obtained from Boston Children's Hospital, as described above. Human lymph node and spleen sections were obtained through the Dana Farber/Harvard Cancer Center Specialized Histopathology Services (DF/HCC SHS) Core. Tonsil, lymph node, and spleen sections were stained by the DF/HCC SHS Core using the following protocol: Antigen retrieval on Lecia Bond, H1(30) (Citrate) for 30 min, followed by primary incubation with Gal-9 antibody (10 μg mL$^{-1}$, Biolegend), PAX5 antibody (CST, 1:50), or isotype control (10 μg mL$^{-1}$, Biolegend) diluted in Leica antibody diluent, for 30 min. Antibodies were detected with either anti-mouse or anti-rabbit secondary antibody-HRP conjugates followed by development with DAB (Leica Bond Refine Detection Kit) and counterstaining with hematoxylin.

For lectin immunohistochemistry, the same procedure was used, except that sections were blocked with Carbo-Free blocking buffer (Vector) and endogenous biotin was blocked with the Streptavidin/Biotin blocking kit (Vector). Staining was performed with STA lectin (2 μg mL$^{-1}$) and detected with streptavidin-HRP conjugate followed by development with DAB (Leica Bond Refine Detection Kit) and counterstaining with hematoxylin.

**Confocal microscopy.** Naive B cells were magnetically enriched from human tonsil by negative selection as described above. Each channel of a poly-L-lysine slide (ibidi) was loaded with 30 μL of a $5 \times 10^6$ mL$^{-1}$ naive B cell suspension with or without treatment reagents added. Cells were treated with or without recombinant human Gal-9 (4 μg mL$^{-1}$) for 15 min in a 37 °C incubator, iced for 3 min, fixed for 30 min with 3% ice-cold paraformaldehyde in PBS, then washed 3× with ice-cold PBS. As a negative control, cells were either left untreated, or treated in the presence of lactose (50 mM).

For staining, cells were permeabilized with 0.1% Tween-20 in PBS for 10 min at room temperature and blocked with 1% BSA, 0.1% Tween-20 in PBS for 20 min at room temperature. Antibody-conjugate stain against CD45 (Abcam) was diluted in 1% BSA, 0.1% Tween-20 in PBS and stained for 30 min at room. Slides were mounted in Prolong Gold with DAPI (CST) and imaged using an Olympus Fluoview FV1000 confocal microscope at the Optical Microscopy Core Facility of the Program in Cellular and Molecular Medicine at Boston Children's Hospital equipped with Fluoview software version 3.1. Images were acquired with a 60X water immersion lens and 3X optical zoom for a total magnification of 180×.

**Immunoprecipitation and co-immunoprecipitation.** GC-depleted B cells or naive B cells were purified by magnetic negative selection techniques as described above. For Gal-9 receptor discovery experiments, cells were first biotinylated to label cell surface glycoproteins using 2 mM EZ-link Sulfo-NHS-biotin (Thermo) in PBS (pH 8.0) followed by quenching in 100 mM glycine. For Gal-9 IP, cells were treated with 2 μg mL$^{-1}$ recombinant Gal-9 (R&D) in 1% BSA for 30 min on ice, followed by washing and lysis in 2% NP-40 buffer/Buffer A (150 mM NaCl, 0.5 mM Tris, 1 mM EDTA). Lysates were quantified using Bradford reagent to ensure equal protein loading. Gal-9 receptors were immunoprecipitated from at least 50 μg lysate (200 μg for endogenous IPs) mixed with 40 μL of a BSA-blocked Protein G-agarose bead slurry (Life Technologies) loaded with anti-Gal-9 antibody (Biolegend) or anti-CD45 antibody (BioRad). Immunoprecipitation was carried out overnight on a rotator, followed by washing with lysis buffer, elution by boiling in Laemmli reducing sample buffer, and western blot with either streptavidin-800CW (Li-Cor) (for detection of biotinylated surface proteins), Gal-9 antibody (R&D), or CD45 antibody (CST). As controls, IPs were performed in parallel with equal amounts of respective isotype control antibody, CD45 antibody, or Gal-9 antibody in the presence of 200 mM lactose to account for non-carbohydrate dependent interactions. For IP reactions against endogenous Gal-9, the procedure was the same, except that no recombinant Gal-9 was added.

**Analysis of BCR signaling by western blot.** Naive B cells were magnetically enriched to >90% purity by negative selection, then rested overnight at 37 °C, 5% CO$_2$ at $2.5 \times 10^6$ cells mL$^{-1}$ in complete Iscove's Modified Dulbecco's Medium (10% FBS, 1% Penicillin/Streptomycin, 15 μg mL$^{-1}$ gentamycin, 25 mM HEPES) supplemented with IL-4 (20 ng mL$^{-1}$, Biolegend). The following morning, the cells were pelleted and resuspended at $5 \times 10^6$ cells mL$^{-1}$ in serum-free IMDM, distributed equally into tubes corresponding with each stimulus, and rested for 1 h at 37 °C, 5% CO$_2$. Stimuli (recombinant human Gal-9 (R&D, 2 μg mL$^{-1}$ or as specified), goat anti-human IgM F(ab')$_2$ (Jackson Immunoresearch, 15 μg mL$^{-1}$), or both were added and the cells were incubated at 37 °C, 5% CO$_2$ for the indicated times. As a negative control of Gal-9 activity, cells were treated in the presence of the competitive inhibitor lactose (25 mM) where indicated. At the end of the time

course, the cells were immediately iced for 1 min, pelleted, and lysed in RIPA buffer (ThermoFisher) supplemented with PMSF and protease/phosphatase inhibitors (Protease/ Phosphatase Inhibitor Mini tablets, ThermoFisher) for a half hour on ice. Lysates were cleared of debris by centrifugation, boiled for 10 min in Laemlli reducing sample buffer, and loaded onto 4–20% Criterion Tris-HCl polyacrylamide gels (BioRad). Gels were transferred to nitrocellulose membranes (0.2 μm pore size) for western blotting. Membranes were blocked in Odyssey Blocking buffer (Li-cor) for 1 h, blotted overnight with primary antibodies against the indicated phospho-proteins, and detected with species-specific IRDye 800CW or 680RD antibody conjugates (Li-Cor) for a half hour. Antibodies were diluted in 1:1 Odyssey Blocking buffer (Li-Cor)/TBS-Tween 20 (0.1%). Blots were scanned using a Li-Cor Odyssey CLx Near-infrared Imaging System. After scanning, blots were stripped with NewBlot IR Stripping buffer (Li-Cor) until signal was eliminated, and reprobed with antibodies against total proteins. Band intensities were calculated using ImageStudio software (Li-Cor), and phospho-protein signals were normalized to the total protein levels of each corresponding analyte. Uncropped western blots are available in Supplementary Figure 9.

**Analysis of BCR calcium flux by flow cytometry.** Tonsil mononuclear cells were thawed, washed, and counted as described above. To block Fc receptors, cells were resuspended in human TruStain FcX Fc blocking antibody (Biolegend) diluted in FACS buffer (2% FBS in PBS) for 10 min on ice. Cells were then stained for surface antigens using antibodies against human CD44 and human CD19 (both Biolegend) for 45 min on ice. Cells were subsequently washed two times with FACS buffer before staining with Fluo-4 No-Wash (NW) Calcium Assay kit (ThermoFisher) according to the manufacturer's instructions. Briefly, cells were resuspended in Assay Buffer at $5 \times 10^6$ cells per mL and incubated at 37 °C for 30 min. Afterwards, an equal volume of Fluo-4 dye was added and the cells were incubated again at 37 °C for 30 min, followed by an additional 30 min incubation at room temperature. A baseline reading was acquired for 30 s, after which a premixed combination of goat anti-human IgM F(ab')$_2$ and donkey anti-human IgG F(ab')$_2$ (each at a final concentration of 20 μg mL$^{-1}$, Jackson Immunoresearch), rhGal-9 (2.5 μg mL$^{-1}$), and/or lactose (25 mM) were added; acquisition was resumed for another 4 min and 30 s. Non-GC B cells were electronically gated (CD19+CD44+), and calcium flux readings were analyzed using the Kinetics platform in FlowJo software (TreeStar). Fluo-4 intensity values were normalized to the average signal intensity of the 30 s baseline. In some experiments, where indicated, anti-IgD F(ab')$_2$ or anti-κ/λ light chain F(ab')$_2$ (final concentration of 20 μg mL$^{-1}$, both Southern Biotech) were used rather than anti-IgM F(ab')$_2$ and anti-IgG F(ab')$_2$.

For assays evaluating endogenous cell surface Gal-9, tonsil mononuclear cells were treated exactly as above, except cells were cultured in Assay buffer containing 25 mM lactose, and anti-BCR stimuli were prepared in Assay buffer supplemented with 25 mM lactose.

For assays involving Karpas 1718 and NUDUL-1 Gal-9 knockdown cell lines, cells were prepared as above, except that the surface stain was omitted. Karpas 1718 B cells were stimulated with goat anti-human IgM F(ab')$_2$ and NUDUL-1 B cells were stimulated with donkey anti-human IgG F(ab')$_2$, respectively, as described above.

**Enzymatic removal of cell surface sialic acids.** To remove cell surface sialic acids, live tonsillar mononuclear cells were washed and resuspended in either serum-free RPMI 1640 alone or RPMI + *Arthrobacter ureafaciens* sialidase (Roche, [final] = 125 mU mL$^{-1}$) and incubated at room temperature for 1 h. Cells were pelleted and washed 3 with PBS before proceeding with calcium flux assessments, as described above. Efficiency of sialic acid removal was measured by flow cytometric staining with *Sambucus nigra* agglutinin and *Maackia amurensis* agglutinin-II, as described above.

**Glycoprotein internalization assay.** Naïve B cells were magnetically enriched from tonsil by negative selection as described above, followed by treatment with recombinant Gal-9 or Gal-9 in the presence of lactose (25 mM) for 0.5, 2, 10, 30, 60, or 120 min at 37 °C, 5% CO$_2$. Gal-9 was added in reverse timecourse order to synchronize cells for antibody staining at the end of the incubation period. Following the incubation, cells were iced, washed with ice-cold buffer, stained with monoclonal antibodies against CD45, CD22, CD19, IgM, IgD, or Gal-9 for 45 min on ice. Following surface staining, the cells were washed again, and then acquired on a FACS Canto I. As a control for steric effects of Gal-9 on antibody binding, a separate group of cells were incubated with Gal-9, but Gal-9 was eluted just prior to antibody staining (during the wash steps) using wash buffer containing 25 mM lactose.

**NFAT1 nuclear translocation assays.** For western blot analysis of NFAT1 nuclear translocation, untouched naïve B cells were enriched by negative selection and rested for 1 h at 37 °C in serum-free RPMI prior to treatment with one or a combination of the following reagents for 30 min at 37 °C: anti-human IgM F(ab')$_2$ (15 μg mL$^{-1}$), recombinant human Gal-9 (2 μg mL$^{-1}$), lactose (25 mM), cyclosporin A (100 ng mL$^{-1}$), or ionomycin (1μM). Cells were subsequently iced for 1 min, pelleted, and lysed as described[77]. Briefly, cells were lysed in cytoplasmic extract (CE) buffer (0.075% NP-40, 10 mM HEPES, 60 mM KCl, 1 mM EDTA,

1 mM β-mercaptoethanol, 1 mM PMSF, protease/phosphatase inhibitor tablet (ThermoFisher)) for 5 min on ice. Lysates were pelleted, CE buffer was removed, and nuclei were lysed with a high-salt nuclear extract (NE) buffer (20 mM Tris, 420 mM NaCl, 1.5 mM MgCl$_2$, 0.2 mM EDTA, 1 mM PMSF, protease/phosphatase inhibitor tablet (ThermoFisher) for 10 min on ice. Lysates were again pelleted for debris and probed by western blot for NFAT1 or loading controls (β-tubulin or Histone H3 for cytoplasmic and nuclear fractions, respectively). Band intensities were calculated using ImageStudio software (Li-Cor), and phospho-protein signals were normalized to the housekeeping protein levels for each fraction.

For detection of NFAT1 translocation by immunofluorescence microscopy, cells were stimulated as described above with the following modifications: cells were layered onto poly-L-lysine coverslips, fixed in situ with 4% paraformaldehyde for 15 min, quenched with 50 mM ammonium chloride, and simultaneously blocked and permeabilized in 5% BSA/0.3% Triton-X100/PBS for 1 h. Cells were stained overnight with NFAT1 antibody (CST, 1:50), detected with donkey-anti-rabbit AlexaFluor 555 secondary antibody (Biolegend) for 30 min, and mounted with ProLong Gold anti-fade reagent with DAPI (CST) overnight. Slides were imaged on a Nikon Eclipse E600 Microscope using a 100x oil immersion objective. For quantification of NFAT1 localization, each cell in a given field was given a binary score of either 1 (predominantly cytoplasmic NFAT1) or −1 (predominantly nuclear NFAT1), and an average score was calculated per field. For each treatment condition, average scores from at least seven fields are reported.

**Statistical analysis.** Statistical analyses were performed using Prism 7.0 software (GraphPad). Appropriate statistical tests were chosen with the assumption of sample normality and equality of variance. For tests involving two groups, hypothesis testing was carried out using an unpaired two-tailed $t$-test. For tests involving three or more groups, a one-way analysis of variance (ANOVA) test was used with Tukey's correction for multiple comparisons. In cases of unequal variance, the Kruskall–Wallis test was used instead. Where appropriate, a one-sample $t$-test was used, where groups were compared to a hypothetical value of 1 or 100. For tests comparing two groups sampled at numerous timepoints, as in calcium assays, a two-way repeated measures ANOVA with Bonferroni correction for multiple comparisons was used against the factors "time" and "treatment." Throughout the study, error bars depict standard error of the mean (SEM) from biological replicates. In each case, sample sizes were estimated based on effect sizes observed in similar studies of glyco-immunology and B cell signaling, including those from our own laboratory. In most cases, the experimenter performed and analyzed the data without blinding, except in analysis of NFAT1 nuclear translocation by immunofluorescence (Fig. 5f, g), where the experimenter was blinded to treatment condition before performing quantification. $P$ values <0.05 were considered statistically significant.

**Data availability.** Datasets generated in this study are either included in this published article or are available from the corresponding author on reasonable request. *LGALS9* expression analysis in human hematopoietic cells was conducted using the GEO dataset GSE24759[26].

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

## Acknowledgements

The authors thank Drs. Steven Barthel, Shiv Pillai, Michael Carroll, Galit Alter, W. Nicholas Haining, and Vini Mani for many helpful discussions, Dr. Francesco Marangoni for assistance with NFAT1 nuclear localization assays, Dr. Gordon Freeman for generously providing recombinant CD40L-trimer, Chad Araneo and John Sullivan at the Harvard Microbiology and Immunobiology Flow Cytometry Core for assistance with FACS sorting, Drs. Jose Martinez-Climent and Alan Epstein for generously providing B cell lines, and the Dana Farber/Harvard Cancer Center Specialized Histopathology Services Core for assistance with immunohistochemistry. Funding: This research was funded by an American Association of Immunologists Careers in Immunology Fellowship (to N.G. and C.J.D.), an Albert J. Ryan foundation fellowship (to N.G.), NIH grants NIH/NIAID R21AI125476 (to C.J.D.) and NIH/NCI R01 CA173610 (to C.J.D.), a Biotechnology and Biological Sciences Research Council grant BBF0083091 (A.D. and S.M.H.) and BBK0161641 (A.D. and S.M.H.), and a Wellcome Trust grant (082098 to A.D.).

## Author contributions

N.G. and C.J.D. conceived the study. N.G., J.L., S.K. and A.A. performed the experiments and analyzed the data. N.G., J.L., J.S., S.K., A.A., A.D., S.M.H., H.R.W. and C.J.D. contributed intellectually to the study. N.B., S.M.P. and G.S.L. assisted with tonsil tissue acquisition. S.M.H. and A.D. supervised MS glycomics assessments. C.J.D. supervised the entire study. N.G., A.A., S.M.H. and C.J.D. wrote the manuscript.

## Additional information

**Competing interests:** The authors declare no competing interests.

