## [Peer Review File · Nature Communications]

Reviewers' comments:

Reviewer #1 (Remarks to the Author):

In this manuscript the authors explore the role of Galectin-9 (Gal-9) in regulation of three human tonsil B cell subsets, namely naïve B cells, memory B cells and germinal center (GC) B cells. This study is of genuine interest with the recent elucidation of the discrete antigen-driven events in the GC that ultimately lead to activation of naïve B cells to enter GCs and undergo differentiation to GC light zone B cells that undergo competitive antigen-dependent selection to enter the long-lived memory B cells and plasma cell pools. The authors provide convincing evidence for a novel finding namely that GC B cells, unlike naïve B cells, do not express the target of Gal-9 binding, namely linear LacNAc but rather express I-branched LacNAc which leaves them resistant to the demonstrated inhibitory effect of Gal-9 binding on naïve B cells.

My questions concern the observation that naïve B cells express Gal-9 protein and which is detectable on their cell surfaces (Fig. 3). Despite endogenous expression of Gal-9, the authors only demonstrate an effect of Gal-9 on naïve B cells when Gal-9 is added exogenously at 2.5 µg/ml. This raises the question of the biological relevance of these findings unless Gal-9 is produced in higher amounts by naïve B cells when activated. The authors show that in the presence of anti-IgM and CpG or CD40L-Tri, CD86 expression on naïve B cells is decreased (Fig. 2). What is the effect of anti-IgM, CpG and/or CD40L-Tri on Gal-9 expression? Does lactose treatment alone alter the level of expression of anti-IgM induced CD86 expression? Similar questions are applicable to the results presented in Fig. 4 and 5. The authors identify CD45 as a ligand for CD45. CD45 has been reported to be spatially organized on B cell surfaces in a non-random fashion. Do the authors observe any effect of lactose treatment of naïve B cell producing Gal-9 on the spatial array of CD45.

Minimally, discussing these issues and modifying Fig. 7 accordingly would improve the manuscript.

Reviewer #2 (Remarks to the Author):

The manuscript of Giovannone et al on Galectin-9 suppresses B cell receptor signaling and is regulated by I-branching of N-glycans describes the interaction of glycoproteins on the surface of B-lymphocytes with the soluble lectin Galectin-9 (Gal-9). Currently, there is an increasing interest in the immunomodulating role of different Galectin family members but most studies are done with T-cells and not B-cells. The authors first analyze the different types of glyogroups on the surface of naive, germinal centers (GC) and memory B-cells. Interestingly, they find that linear polymeric glyogroups are mostly found on naive and memory B-cells whereas GC B-cells frequently have I-branched glyogroups. The authors point out that, in contrast to galactin-1, Gal-9 is specific for the linear polymeric glyogroups and does not bind the I-branched glyogroups. Indeed, in a FACScan analysis they show convincingly that Gal-9 binds to naive and memory B-cells but not GC B-cells. In addition, they show in a tissue staining study that Gal-9 is only poorly staining GC structures. In line with this the authors found that in comparison to naive follicular B-cells, the GC B-cells express more of the glycosyltransferase GCNT2, the enzyme conducting the I-branching of poly-LacNAc structures. They suggest that the GCNT2 expression makes GC B-cells prone to BCR signal inhibition by Gal-9. In a functional study they show that exposure of B-cells to Gal-9 inhibits the anti-IgM + CpG or anti-IgM + CD40L induced proliferation. In line with this they found that Gal-9 is reducing the anti-IgM induced Calcium mobilization and NFAT activity. The authors then study the interactions of Gal-9 with the B-cell surface. They use biotinylated naive B-cells to detect Gal-9 bound protein and identify the protein tyrosine phosphatase CD45 as the binding target of this lectin. They then show by Westernblotting employing either a general (4G10) or site-specific anti-phosphotyrosine antibodies that the exposure of B-cells to Gal-9 results in an increased tyrosine

phosphorylation of the Src-family kinase Lyn and the inhibitory receptor CD22. They suggest that the exposure of Gal-9 to B-cells promotes interaction of CD45, Lyn and CD22 thus promoting CD22 tyrosine phosphorylation by Lyn, the recruitment of the tyrosine phosphatase SHP1 and inhibition of BCR signaling.

This manuscript provides novel insights in the regulation of B-lymphocytes and is as such suitable for publication in Nature Communication. Most studies of this manuscript are well done and I have only three major comments:

- 1.) in figure 5 an important control is missing namely the exposure of the B-cells to Gal-9 alone using 2.5 µg/ml of Gal-9. By the way, in this assay the authors mark the B-cells with an anti-CD19 antibody and they should mention this as it is well known that anti-CD19 antibodies can augment the calcium response of anti-BCR stimulated B-cells.
- 2.) If the author could show that only in the presence (but not in the absence) of Gal-9 they can co-precipitate CD22 with anti-CD45 antibodies or vice versa CD45 with anti-CD22 antibodies this would support their model that Gal-9 is promoting a CD45/CD20 interaction.
- 3.) The authors should conduct a FACScan analysis showing that the exposure of human tonsillar B cells to 2.5µg/mL of Gal-9 is not changing the expression of IgM, IgD and CD19 on the surface of these B cells.

Minor comments:

The term "We magnetically sorted untouched naïve B cells from tonsil" is not correctly representing the used experimental procedure. The authors should rephrase the sentence by stating clearly that they did not sort but enriched for B cells by depleting other mononuclear cells. The sentence on line 173 contains twice "detectable"

Reviewer #3 (Remarks to the Author):

This manuscript describes a novel role for the extracellular lectin, Galectin-9 (Gal-9), in regulating BCR-induced Ca²⁺ signaling and the activation of human B cells. By comparing the "glycome" of human naïve, germinal center (GC), and memory B cells, and combining this with flow cytometry using a variety of plant lectins, the authors show that naïve and memory B cells express Gal-9-binding carbohydrate moieties on their surface and are subject to negative regulation by Gal-9. In contrast, GC B cells modify their carbohydrate structures with I-branched N-glycans such that Gal-9 binding is greatly reduced and, consequently, BCR signaling and subsequent activation events are resistant to the inhibitory effects of Gal-9. They go on to show that expression of the glycosyltransferase GCNT2 was upregulated in GC B cells, and that this is necessary and sufficient to block Gal-9 binding. The authors also show that B cells produce Gal-9 (although they can also bind exogenous Gal-9, which is made by a number of other cell types) and that Gal-9 production is downregulated in GC B cells compared to naïve B cells. This suggests that Gal-9 differentially tunes the threshold for B cell activation in naïve B cells, where it limits B cell activation by low amounts of antigen or perhaps self-antigen. In contrast, GC B cells would not be subject to this negative regulation and may have a lower threshold for activation, with implications for both protective antibody responses and autoimmunity. These are all novel and significant findings that are strongly supported by high quality data. The unique and sophisticated glycomics analysis, as well as the use of primary human B cell subsets, are significant strengths of this manuscript. One significant concern that the authors should address is how the concentrations of exogenous Gal-9 required to dampen BCR signaling and B cell activation compare to concentrations that B cells might encounter in vivo.

As a possible mechanism for the inhibition of BCR signaling by Gal-9, the authors show that Gal-9 binds to CD45 and that the binding of Gal-9 to B cells increases the activity of the CD22/Lyn/SHP-1 pathway (as judged by phosphorylation of these 3 proteins on key residues), which is known to limit BCR signaling. The addition of Gal-9 to naïve B cells results in capping of CD45, and in the

presence of anti-IgM, the authors suggest that IgM-BCRs, CD45, and CD22 all co-localize or co-cap. However, these images are of low resolution, the significance of "capping" is unclear, and the authors' proposal that Gal-9 promotes the co-localization of CD22 and the BCR in Lyn-containing signaling domains (Fig. 7) is not supported by the data. Higher magnification microscopy images as well as higher resolution techniques such as proximity-ligation assays or super-resolution microscopy would be needed to support this model. Moreover, no functional tests are done to support the idea that engaging the CD22/Lyn/SHP-1 pathway represents an important mechanism by which Gal-9 dampens BCR signaling. This could be tested by using siRNA or shRNA (as in Fig. 6) to knock down the expression of CD22 or SHP-1 in Ramos cells. This section of the manuscript is the least convincing and could be strengthened.

Main points for the authors to address:

1. A significant concern is how the concentrations of exogenous Gal-9 required to dampen BCR signaling and B cell activation in vitro (1-4 $\mu\text{g/ml}$; 25-100 nM) compare to the local Gal-9 concentrations that B cells might encounter in vivo. Although there may be little information about this, if the Gal-9 concentrations used in this study are much higher than what a B cell might reasonably encounter in vivo, this brings into question the physiological significance of these findings. At the very least, the authors should discuss whether similar concentrations of exogenous Gal-9 have been used in other studies examining the effects of Gal-9 on lymphocytes.

a) The flow cytometry data in Fig. 3e show that the amount of Gal-9 on the surface of ex vivo B cells has an MFI of 10-50, instead of an MFI of 4000 when the cells were incubated with 1 $\mu\text{g/ml}$ (25 nM) recombinant Gal-9 (25 nM) in vitro (graphs in Fig. 1a). Is the affinity/avidity of Gal-9 binding so low that as soon as the cells are exposed to Gal-9-free medium during the isolation process, most of in vivo bound Gal-9 dissociates from the B cell surface? If that's not the case, then one could argue that the amount of exogenous Gal-9 being added to the cells in vitro results in 100 times more Gal-9 bound to the surface of the B cell than what they are normally have on their surface in vivo.

b) Does adding lactose to ex vivo naïve B cells result in increased BCR signaling? This would address whether the amount of in vivo-derived Gal-9 bound to the ex vivo B cells is sufficient to limit BCR signaling. I don't think I saw any lactose-only controls in the absence of Gal-9 addition.

2. Some additional controls for the BCR signaling experiments are needed. The authors should show by FACS that addition of exogenous Gal-9 does not reduce the binding of anti-IgM to cell surface BCRs. This would eliminate a trivial explanation for why exogenous Gal-9 impairs anti-IgM-induced B cell activation. They should also test whether Gal-9 addition alters BCR internalization. For example, if Gal-9 promotes BCR internalization this could reduce the level of BCR on the cell surface, resulting in less BCR signaling in response to anti-IgM.

3. In Fig. 3g, the authors show that when exogenous Gal-9 is added to naïve B cells they can co-immunoprecipitate Gal-9 and CD45. Since they implicate Gal-9 in regulating the signaling functions of membrane IgM and CD22, they should repeat this experiment to determine if IgM and CD22 also co-precipitate with Gal-9. If so, it would support their model that Gal-9 brings CD45 and CD22 into close proximity with (in the same signaling domain) as the BCR.

4. In Fig. 4, phosphoprotein band intensities are normalized to the loading control but on the graphs they appear to also have been normalized to the 0 anti-IgM/0 Gal-9 sample, which is set to 1.0 on all of the graphs. If the 0 anti-IgM and + anti-IgM samples are on different blots, I don't think one can directly compare these values to each other. Only samples from the same blot can be quantified relative to each other. The authors should clarify this.

a) Images of the full blots should be shown for all experiments, as per journal policy.

5. Fig. 5: Naïve B cells express both IgM- and IgD-containing BCRs. Does Gal-9 also inhibit BCR signaling induced by anti-IgD antibodies? Does it inhibit signaling induced by anti-Ig light chain antibodies? This would mimic antigen stimulation by binding to both types of BCRs. If Gal-9 inhibited IgM-BCR signaling but not IgD-BCR signaling, this would be very interesting and suggest perhaps that Gal-9 mediates some of its effects by binding directly to IgM (as suggested by the accompanying manuscript).

6. Fig. 5 and Fig. S4: The authors should discuss possible reasons why Gal-9 inhibits BCR-induced Ca²⁺ signaling but not other aspects of BCR signaling.

a) The CD22/Lyn/SHP-1 pathway has been shown to inhibit BCR-induced phosphorylation of BLNK, BTK, and PLC-gamma, which constitute a central Ca²⁺ signaling module. Showing that Gal-9 inhibited the phosphorylation of these proteins would provide more mechanistic insights and argue against alternative explanations, e.g. Gal-9 affecting the function of plasma membrane Ca²⁺ channels.

b) Alternatively, anti-IgM concentrations required for BCR-induced Ca²⁺ signaling may be higher than for other signaling responses, and hence more sensitive to modest inhibition of proximal BCR signaling events. Experiments using sub-optimal (i.e. less than saturating) concentrations of anti-IgM may reveal that other BCR signaling pathways are also modulated by Gal-9. This type of experiment would fit with the idea that Gal-9 tunes the threshold for B cell activation.

c) In Fig. S4, Gal-9 alone causes a 4- to 10- increase in pAkt and pERK that is reversible by lactose. The authors should discuss whether this represents an increase in basal BCR signaling caused by Gal-9.

d) In Fig. 5f it would be good to add a control to show that exogenous Gal-9 does not inhibit ionomycin-induced nuclear translocation of NFAT1.

e) Fig. 5d-f. Immunofluorescence (IF) is a much better technique for quantifying nuclear translocation than cell fractionation, which has many caveats. The IF data in Fig. 5f should be quantified as percent of cells with NFAT in the nucleus.

7. Fig. S5. The authors show that treating B cells with anti-IgM + Gal-9 causes co-capping of IgM, CD45, and pCD22. From these data, they argue that this supports a model in which CD45 brings CD22 close to the BCR and thereby inhibits BCR signaling. The physiological relevance of "capping" is not clear, except maybe for antigen internalization, and this response occurs at much later times (15 min in the experiment) than the Gal-9-mediated inhibition of BCR-induced Ca²⁺ signaling, which occurs in less than 1 min. As well, controls are needed to show that Gal-9 doesn't cause other membrane proteins to cap. Would Gal-9 cause all membrane proteins with poly-lactosamines to cap and become polarized along with the BCR? It would be good to assess the specificity of this response.

a) These images are of low magnification and resolution and do not support the authors' proposal that Gal-9 promotes the co-localization of CD22 and the BCR in Lyn-containing signaling domains (Fig. 7). Higher magnification microscopy images as well as higher resolution techniques such as proximity-ligation assays or super-resolution microscopy would be needed to support this model.

8. Fig. 6. The experimental results using Gal-9 +/- lactose could be strengthened by repeating a few key signaling experiments in the paired cell lines with and without GCB-like I-branched membrane proteins. Because the cell lines are identical except for GCNT2 expression, this would reveal whether I-branching affects BCR signaling by itself and be an excellent direct test of whether I-branching renders B cells resistant to the effects of Gal-9.

9. Fig. 7. No functional evidence is presented to support the idea that Gal-9 inhibits BCR signaling largely by acting via CD22/Lyn/SHP1 pathway. This should be tested by using siRNA or shRNA (as in Fig. 6) to knock down the expression of CD22 or SHP-1 in Ramos cells and seeing if this largely ablates the inhibitory effects of Gal-9 on BCR signaling. This is a critical experiment to support the authors' signaling data and model.

Other points for the authors to address:

1. The legend to Fig. 1a should indicate what each of the colors represents – naïve, GC, and memory B cells?
2. Figs. 1c, 1e and Fig. S1. For the glycome analysis, the figure legends should provide information about reproducibility, specifically how many times the same analysis was performed on samples from different individuals, and what the variability between samples was.
3. Fig. 4a should be moved to supplemental data as the identification of these 4G10-reactive bands is not definitive and Fig. 4b directly tests the hypothesis that they are the phosphorylated forms of CD22 and Lyn
4. There is a mistake in legend to Fig. 5. CsA and iono were used in Fig. 5d, but 3 lines from the end of the legend it says CsA and PMA (instead of iono).

Point-by-Point Responses to the Referees' Comments to NCOMMS-17-32912-T:

REVIEWER #1

Major Point 1) The reviewer expresses the following concern: “*Despite endogenous expression of Gal-9, the authors only demonstrate an effect of Gal-9 on naïve B cells when Gal-9 is added exogenously at 2.5 µg/ml. This raises the question of the biological relevance of these findings... Does lactose treatment alone alter the level of expression of anti-IgM induced CD86 expression?*”

Our Response: We share the reviewer’s concern about the potential relevance of exogenous Gal-9 in functional assays. We therefore sought to bolster these findings with experiments assaying B cell-intrinsic Gal-9, via two complementary approaches:

- i. Manipulation of pre-existing, surface-bound Gal-9 on primary B cells through lactose-mediated elution
- ii. Generation of stable Gal-9 knockdown (KD) B cell lines via lentiviral transduction of Gal-9 shRNAs

These new findings are presented in **Fig. 7**. Based on our existing data that exogenous Gal-9 suppresses B cell receptor-mediated calcium flux, we expected that inhibition of Gal-9 function by lactose elution of surface Gal-9 or genetic knockdown would result in enhanced BCR calcium flux. As expected, we found that inclusion of lactose alone was sufficient to enhance calcium flux in primary B cells (**Fig. 7a**). Moreover, in a Gal-9^{hi} CD22⁺ SHP1⁺ B cell line (Karpas 1718), but not in a Gal-9^{hi} CD22^{lo} SHP1^{lo} cell line (NUDUL-1), knockdown of Gal-9 resulted in augmented peak, and total calcium flux that was reversed by addition of recombinant Gal-9 (**Fig. 7c-f**). Taken together, these data complement our findings with recombinant Gal-9 and support our proposed model of Gal-9-mediated regulation of B cell receptor calcium signaling.

Major Point 2) The reviewer expresses the following concern: “*The authors show that in the presence of anti-IgM and CpG or CD40L-Tri [plus Gal-9], CD86 expression on naïve B cells is decreased (Fig. 2). What is the effect of anti-IgM, CpG and/or CD40L-Tri on Gal-9 expression?*”

Our Response: We were also interested in determining whether B cell activation impacts Gal-9 expression. To this end, we assessed Gal-9 transcript and protein levels in magnetically enriched naïve B cells following T-independent stimulus, T-dependent stimulus, or no stimulus. Intriguingly, Gal-9 expression showed rapid decline in *ex vivo* cell culture even without stimulation, therefore complicating interpretation of results in the presence of T-independent or T-dependent activation stimuli (presented on next page). Nonetheless, activation *in vitro* does not appear to be sufficient to upregulate Gal-9 expression.

The downregulation of Gal-9 *in vitro* suggests that environmental cues may be necessary to maintain Gal-9 expression *in vivo*. Because we found that naïve B cells in peripheral blood also express very high levels of Gal-9 (**Supplementary Figure 5c**), we do not believe that Gal-9 expression is unique to tonsil-localized B cells. How Gal-9 expression levels in B cells may be extrinsically regulated *in vivo* is currently being investigated by our laboratory.

Figure corresponding with Reviewer 1 Major Point 2

Galectin-9 is downregulated upon *in vitro* B cell culture. Untouched naïve B cells were enriched by MACS before *in vitro* stimulation with either T-independent or T-dependent stimuli, as in **Fig. 6c**. For no stimulation condition, cells were cultured in IL-4 alone (20ng/mL). Cells cultured in the absence of cytokine displayed similar results to IL-4 alone (not depicted). **(a)** qRT-PCR of *LGALS9* gene expression following *in vitro* naïve B cell stimulation, normalized to housekeeping gene *VPS29* and presented relative to 0hr timepoint in each case. **(b)** Representative western blot of Gal-9 expression following *in vitro* naïve B cell stimulation. **(c)** Quantification of Gal-9 expression following *in vitro* naïve B cell stimulation. For (a) and (c), n=2 tonsil specimens over two independent experiments. Statistics were calculated using one-way ANOVA corrected for multiple comparisons (a) or a one-sample T-test against a hypothetical value of 1.0 (b). Error bars represent SEM. *p<0.05.

Major Point 3) The reviewer expresses the following concern: “*The authors identify CD45 as a ligand for CD45. CD45 has been reported to be spatially organized on B cell surfaces in a non-random fashion. Do the authors observe any effect of lactose treatment of naïve B cell producing Gal-9 on the spatial array of CD45?*”

Our Response: We agree with the reviewer that assessing the spatial distribution of CD45 would be of great interest, especially given established roles for galectins in regulating the membrane partitioning of glycoproteins¹. Though we observe gross alterations in CD45 membrane localization by confocal microscopy following Gal-9 treatment (**Fig. 3f**), super-resolution microscopy techniques would likely be required to reliably assess more subtle effects on CD45 membrane distribution, as might be expected following perturbation of endogenous Gal-9. Indeed, in murine B cells, CD45 and CD22 are known to cluster in nanodomains that are best observed using techniques such as dSTORM^{2,3}, as implemented by Dr. Treanor and colleagues in the accompanying manuscript. Given the level of technical expertise required for these techniques, we plan to pursue these studies on human B cells in a subsequent manuscript.

Major Point 4) The reviewer states that, “Minimally, discussing these 3 issues and modifying Fig. 7 accordingly would improve the manuscript.”

Our Response: We thank the reviewer for the suggestion. We have modified our model (now **Fig. 8**) to be less speculative, removing references to membrane re-distribution of CD45. We have also added additional commentary to the Discussion to address the above points.

REVIEWER #2

Major Point 1.) The reviewer expresses the following concern: “*In figure 5 an important control is missing namely the exposure of the B-cells to Gal-9 alone using 2.5 µg/ml of Gal-9. By the way, in this assay the authors mark the B-cells with an anti-CD19 antibody and they should mention this as it is well known that anti-CD19 antibodies can augment the calcium response of anti-BCR stimulated B-cells.*”

Our Response: We agree with the reviewer that treatment of B cells with Gal-9 alone is an important control. As suggested, we conducted the experiment and found that Gal-9 alone was insufficient to induce calcium mobilization (**Supplementary Fig. S7c**).

We also concur with the reviewer that anti-CD19 antibodies may have been influencing the calcium response to anti-BCR and Gal-9-induced flux. We now include data from unlabeled MACS-enriched naïve B cells (negative selection) and observe similar results (see **Supplementary Fig. 7b**). We have also added text to clearly indicate that cells are first labeled with CD19 and CD44 antibodies for gating purposes (see **Fig. 5** legend and **Materials and Methods**).

Major Point 2) The reviewer recommends that “*If the author could show that, only in the presence (but not in the absence) of Gal-9, they can co-precipitate CD22 with anti-CD45 antibodies or vice versa CD45 with anti-CD22 antibodies this would support their model that Gal-9 is promoting a CD45/CD20 interaction.*”

Our Response: We thank the reviewer for this suggestion. In our initial manuscript submission, we proposed a model in which Gal-9 redistributes CD45 to specialized membrane domains containing Lyn, such as lipid rafts, where Lyn has been shown to be enriched. Because CD22 has been reported to associate with CD45 at steady-state^{3,4}, we speculated that Gal-9-mediated re-localization of CD45 may also induce re-localization of CD22 to Lyn-containing domains, in a similar fashion to that reported for T cells⁵. We reasoned that co-localization of CD45, Lyn, and CD22 might thus allow for efficient CD45-mediated activation of Lyn and phosphorylation of CD22 ITIMs. However, we acknowledge that this model was largely speculative and should be reserved for the discussion only (and not the main figures). Therefore, we have modified our model in **Fig. 8** accordingly. Nevertheless, we also sought to further test our hypothesis of a CD22-dependent mechanism of action for Gal-9.

As mentioned above, previous reports have shown that CD22 and CD45 associate at steady-state via the ligand binding domain of CD22 (which binds sialic acids) and the ectodomain of CD45 (which is enriched for sialic acids)^{3,4}. Enzymatic removal of sialic acids is also known to “release” CD22 and cause phosphorylation of CD22 ITIMs, a phenomenon known as CD22 “unmasking”^{6,7}. We therefore reasoned that if Gal-9-mediated suppression depends on activation of CD22, preemptive unmasking of CD22 should abrogate the ability of Gal-9 to suppress calcium flux. To test this, we sialidase treated primary tonsil cells to disengage CD22 from its ligands *in cis*, including CD45 (**Supplementary Fig. 7d**). Although sialidase-mediated unmasking of CD22 dampened overall calcium flux in response to BCR crosslinking, this effect was not absolute and some BCR calcium flux was still observable (**Fig. 5h**, left panel). When we subsequently tested the effect of Gal-9 in sialidase-treated cells, we found that, in support of our hypothesis, Gal-9 was completely unable to suppress BCR-mediated calcium mobilization (**Fig. 5h**, middle and right panels). Importantly, this effect was not due to altered Gal-9 binding activity, as Gal-9 bound equally to sialidase treated and untreated cells

(Supplementary Fig. 7e). In additional experiments, we also examined the effect of Gal-9 knockdown on calcium flux in cell lines expressing measurable levels of CD22 and SHP-1 and cell lines expressing much lower levels of these two molecules (**Fig. 7b**). We found that Gal-9 knockdown was only able to influence calcium flux in the CD22+ and SHP-1+ cell line, but not a cell line lacking meaningful expression of CD22 and SHP-1 (**Fig. 7c-j**). Therefore, although we were not able to independently confirm reports of CD45/CD22 co-immunoprecipitation, these data support the hypothesis that Gal-9-mediated suppression of BCR calcium mobilization is CD22-dependent.

Major Point 3) The reviewer suggests that “*The authors should conduct a FACScan analysis showing that the exposure of human tonsillar B cells to 2.5ug/mL of Gal-9 is not changing the expression of IgM, IgD and CD19 on the surface of these B cells.*”

Our Response: We agree that this is an excellent suggestion, as galectins have been reported to regulate receptor internalization¹. To this end, we now present flow cytometry data evaluating the effect of Gal-9 on the internalization of numerous glycoproteins, including IgM, IgD, CD19, CD45, and CD22. After controlling for steric blockade of antibody binding by Gal-9 (via lactose elution just prior to antibody staining), we observed minimal to no Gal-9-induced internalization (**Supplementary Fig. 8a**). However, without lactose elution, we did find that Gal-9 was capable of blocking Ab binding to CD45 and CD22. By contrast, Gal-9 failed to block binding of Abs to IgM, IgD, and CD19, and in some cases even appeared to augment Ab binding. Therefore, we conclude that Gal-9-mediated inhibition of BCR signaling is not dependent on internalization of IgM, IgD, CD19, CD22, or CD45.

Minor Point 1) The reviewer states that, “*The term, ‘We magnetically sorted untouched naïve B cells from tonsil’ is not correctly representing the used experimental procedure*” and suggests that “*The authors should rephrase the sentence by stating clearly that they did not sort but enriched for B cells by depleting other mononuclear cells.*”

Our Response: We appreciate this clarification and have changed the language accordingly throughout the manuscript.

Minor Point 2) The reviewer states that “*The sentence on line 173 contains twice “detectable”.*”

Our Response: We thank the reviewer and have removed the duplicate word.

REVIEWER #3

Major Point 1) The reviewer expresses several important and related concerns regarding the potential physiological significance of using recombinant Gal-9:

a) *“A significant concern is how the concentrations of exogenous Gal-9 required to dampen BCR signaling and B cell activation in vitro (1-4 µg/ml; 25-100 nM) compare to the local Gal-9 concentrations that B cells might encounter in vivo. ...[I]f the Gal-9 concentrations used in this study are much higher than what a B cell might reasonably encounter in vivo, this brings into question the physiological significance of these findings.*”

b.) *The flow cytometry data in Fig. 3e show that the amount of Gal-9 on the surface of ex vivo B cells has an MFI of 10-50, instead of an MFI of 4000 when the cells were incubated with 1 µg/ml (25 nM) recombinant Gal-9 (25 nM) in vitro (graphs in Fig. 1a). Is the affinity/avidity of Gal-9 binding so low that as soon as the cells are exposed to Gal-9-free medium during the isolation process, most of in vivo bound Gal-9 dissociates from the B cell surface? If that’s not the case, then one could argue that the amount of exogenous Gal-9 being added to the cells in vitro results in 100 times more Gal-9 bound to the surface of the B cell than what they are normally have on their surface in vivo.”*

c.) *“Does adding lactose to ex vivo naïve B cells result in increased BCR signaling? This would address whether the amount of in vivo-derived Gal-9 bound to the ex vivo B cells is sufficient to limit BCR signaling. I don’t think I saw any lactose-only controls in the absence of Gal-9 addition.”*

d.) At the very least, the authors should discuss whether similar concentrations of exogenous Gal-9 have been used in other studies examining the effects of Gal-9 on lymphocytes.”

Our Response. We agree with the reviewer that these are all excellent points to consider. We propose that the concentration of exogenous Gal-9 used (2.5µg/mL) is justified for the following reasons:

- i. We initially examined concentrations of Gal-9 used in similar studies in T cells and found that our use of 2.5µg/mL is consistent with those studies. For example, a study examining the effect of human Gal-9 on Th2-polarized T cells utilized a concentration of 0.1 µM (3.4µg/mL)⁸, whereas another study in Jurkat T cells noted efficacy between 0.1 µM-1µM Gal-9 (3.4µg/mL - 34 µg/mL)⁹. The manufacturer of the recombinant Gal-9 we use in this study (R&D, cat# 2045-GA) also lists the ED₅₀ for their preparation between 1-5µg/mL.
- ii. In our own studies on B cells, saturation binding curve analysis of Gal-9 revealed that 2.5 µg/mL is sub-saturating on naïve B cells and close to the K_d (K_d = 84 +/- 28 nM ~ 2.8 +/- 0.95 µg/mL Gal-9) (**Supplementary Fig. 4a**)
- iii. In the study by Dr. Treanor and colleagues (submitted jointly with this manuscript), the use of Gal-9 null mice allowed for an estimate of the surface concentration of Gal-9 to be approximately 0.1µM (3.4µg/mL) which is in line with the level of Gal-9 used in this study.

However, as the reviewer rightly points out, determination of the precise concentration of galectin-9 encountered by B cells *in vivo* is difficult to ascertain. In part, the difficulty arises

from possible differences in absolute concentrations of Gal-9 in lymphoid tissue / blood vs. effective concentrations of Gal-9 experienced by B cells in their local microenvironment. In this regard, there are several points to consider: (1) B cells residing in follicles are surrounded by other B cells producing Gal-9 (**Fig. 3a and Supplementary Fig 5a,b**), likely increasing the local concentration of Gal-9; (2) The effective concentration experienced by B cells in the follicles may be higher than what may be measured in tissue extracts due to preferential retention of Gal-9 on B cell surfaces (due to expression of high affinity Gal-9 binding determinants, poly-LacNAc); (3) Presentation of Gal-9 on a fixed substrate, such as the membrane of a neighboring B cell, may lower the amount of Gal-9 required to achieve a physiological effect compared to Gal-9 acting in solution. Therefore, in a similar way to other receptor-ligand interactions modeled *in vitro*, administration of monomeric Gal-9 in solution may require much higher concentrations to achieve a functional outcome than when administered as a multimer or fixed to a solid substrate, such as a bead. This is particularly relevant for galectins, which are thought to function by inducing oligomerization of glycoproteins into galectin-glycoprotein “lattices”¹. A similar phenomenon is observed with antibodies in solution vs. bead/ plate-bound in the activation of T cells. Indeed, in the present manuscript, stimulation of calcium flux via soluble F(ab')₂ required as much as 20µg/mL anti-IgM F(ab')₂.

Nonetheless, we also reasoned that directly testing the role of endogenous Gal-9 would circumvent the difficulties of determining physiologically relevant concentrations of exogenous Gal-9. Therefore, we also analyzed the effect of perturbing B cell-intrinsic Gal-9 via the following two disparate methodologies:

- i. Lactose-mediated elution of Gal-9 from primary B cells, as suggested by the reviewer. A strength of this approach is the use of primary B cells, but a limitation is that lactose is theoretically capable of eluting any and all bound galectins.
- ii. Stable knockdown of Gal-9 in a representative Gal-9^{hi}, CD22^{hi}, SHP1^{hi} B cell line (Karpas 1718). A strength of this approach is the specific knockdown of Gal-9, but a drawback is the use of B cell lines rather than primary B cells.

Because our previous results with exogenous Gal-9 suggested that Gal-9 could inhibit BCR signaling, we hypothesized that elution and/or knockdown of Gal-9 would augment BCR calcium flux. The results are presented in **Fig. 7**. Indeed, lactose treatment of primary B cells consistently led to enhanced calcium flux, supporting a constitutive inhibitory role for endogenous Gal-9 on BCR signaling (**Fig. 7a**). Similarly, knockdown of B cell-intrinsic Gal-9 in Karpas 1718 B cells elicited higher peak and total calcium mobilization in response to BCR crosslinking (**Fig. 7c,e-f, Supplementary Fig. 8b**). We also observed a trend toward elevated intracellular calcium concentrations at baseline (**Fig. 7d**), suggesting that Gal-9 may act constitutively to suppress calcium levels. Importantly, addition of exogenous Gal-9 restored peak and total calcium flux to the levels observed in control cells (**Fig. 7k-m**). Taken together with our existing results using exogenous Gal-9, these new data strongly support a role for Gal-9 in dampening BCR signaling.

Major Point 2.) The reviewer recommends that “*Some additional controls for the BCR signaling experiments are needed. The authors should show by FACS that addition of exogenous Gal-9 does not reduce the binding of anti-IgM to cell surface BCRs. This would eliminate a trivial explanation for why exogenous Gal-9 impairs anti-IgM-induced B cell activation. They should also test whether Gal-9 addition alters BCR internalization. For example, if Gal-9 promotes BCR internalization this could reduce the level of BCR on the cell surface, resulting in less BCR signaling in response to anti-IgM.*”

Our Response. We thank the reviewer for suggesting these important control experiments. As recommended, we compared binding of the polyclonal anti-IgM crosslinking F(ab')₂ in the presence or absence of 2.5µg/mL Gal-9 and observed no difference in anti-IgM binding (**Supplementary Fig. 7a**), making steric competition between Gal-9 and IgM F(ab')₂ an unlikely explanation for our findings.

We also utilized flow cytometry to test whether Gal-9 could induce the internalization of IgM, IgD, and several other glycoproteins, including CD45 and CD22. To parse steric competition for mAb binding by Gal-9 vs. bona fide receptor internalization, we also included a “lactose elution” control condition in which Gal-9 was eluted from the surface of B cells immediately before Ab staining. By this approach, we observed no internalization of IgM, IgD, or CD19, and only very minimal internalization of CD45 and CD22 over the two-hour incubation period examined (**Supplementary Fig. 8a**).

Though we did not observe meaningful internalization, we did note significant competition for mAb binding with CD45 and CD22 (**Supplementary Fig. 8a**). For CD45, inhibited mAb binding was not unexpected in light of our finding that CD45 is a receptor for Gal-9 (**Fig. 3e,f; Supplementary Fig. 5d,e**). However, for CD22, it is unclear whether reduced mAb binding resulted from indirect blockade due to association with heavily glycosylated and bulky CD45, or a more direct effect resulting from Gal-9 binding to CD22. We postulate that indirect blockade is the more likely mechanism considering the lower magnitude of inhibition observed with CD22 vs. CD45 and our failure to detect CD22 in our unbiased Gal-9 IP (**Supplementary Fig. 5d**) and direct Co-IP assessments (data not shown).

Collectively, these results suggest that the suppressive mechanism of Gal-9 is not the result of competition with stimulation reagents, nor due to inadvertent or intentional internalization of BCR or related glycoproteins.

Major Point 3) The reviewer makes the following recommendation: “*In Fig. 3g, the authors show that when exogenous Gal-9 is added to naïve B cells they can co-immunoprecipitate Gal-9 and CD45. Since they implicate Gal-9 in regulating the signaling functions of membrane IgM and CD22, they should repeat this experiment to determine if IgM and CD22 also co-precipitate with Gal-9. If so, it would support their model that Gal-9 brings CD45 and CD22 into close proximity with (in the same signaling domain) as the BCR.*”

Our Response. While CD45 and CD22 have previously been reported to associate at steady state^{3,4,7}, we acknowledge that our proposed model of Gal-9-induced CD22 / CD45 / IgM colocalization was largely speculative. Because we were unable to detect CD22 and/or IgM in Gal-9 immunoprecipitates (**Supplementary Fig. 5d** and data not shown), we have altered the model accordingly to remove unwarranted speculation about whether Gal-9 can induce co-localization of CD45 / CD22 with IgM (**Fig. 8**). However, our inability to detect association by IP does not necessarily argue against such a model for two reasons:

- i. Depending on the strength of possible association between CD45, CD22, and/or IgM, our IP conditions may not be optimal. In this manuscript, we used a moderately strong IP buffer formulation that includes 2% NP-40, which may disrupt weaker protein-protein or protein-glycan interactions. In subsequent studies, varying the IP conditions will be important to detect possible complexing between CD45, CD22, or IgM after Gal-9 treatment.
- ii. Rather than directly complex, CD45, CD22, and IgM may be re-distributed by Gal-9 to the same membrane microdomain, such as lipid raft domains,

without strong association. Indeed, whereas Lyn has been shown to be lipid raft resident, CD45 and CD22 are thought to be predominantly excluded from rafts at steady state¹⁰. Indeed, studies in T cells have found that galectins can regulate the localization of CD45 to raft domains⁵. Moreover, results from Dr. Treanor and colleagues (in the accompanying manuscript) in murine B cells / murine Gal-9 support this hypothesis. Therefore, lipid raft isolation and analysis may be a more suitable methodology than IP for assessing CD45 / CD22 / IgM membrane co-compartmentalization, in the absence of direct protein-protein interactions. Our lab is currently investigating the effect of Gal-9 on CD45 and CD22 recruitment to raft domains in human cells.

Major Point 4) The reviewer makes the following recommendation: *“In Fig. 4, phosphoprotein band intensities are normalized to the loading control but on the graphs they appear to also have been normalized to the 0 anti-IgM/0 Gal-9 sample, which is set to 1.0 on all of the graphs. If the 0 anti-IgM and + anti-IgM samples are on different blots, I don’t think one can directly compare these values to each other. Only samples from the same blot can be quantified relative to each other. The authors should clarify this...[Additionally], Images of the full blots should be shown for all experiments, as per journal policy.”*

Our Response. We thank the reviewer for pointing out this omission. We have now have added text clarifying that the images are quantified from the same blot; the images were cropped to remove the molecular weight bands in between the two groups. The full blots are now also included in this submission.

Major Point 5) The reviewer makes the following recommendation: *“Naïve B cells express both IgM- and IgD-containing BCRs. Does Gal-9 also inhibit BCR signaling induced by anti-IgD antibodies? Does it inhibit signaling induced by anti-Ig light chain antibodies? This would mimic antigen stimulation by binding to both types of BCRs. If Gal-9 inhibited IgM-BCR signaling but not IgD-BCR signaling, this would be very interesting and suggest perhaps that Gal-9 mediates some of its effects by binding directly to IgM (as suggested by the accompanying manuscript).”*

Our Response. We agree that this is a very interesting question. We repeated both longer-term activation assays (**Supplementary Fig. 7g**) and shorter-term calcium assays (**Supplementary Fig. 7b**) using all three modes of stimulation – anti-IgM F(ab')₂, anti-IgD F(ab')₂, and combined anti-light chain (Kappa + Lambda) F(ab')₂. We found that Gal-9 was equally capable of suppressing activation / signaling in all three cases, suggesting that, in humans, Gal-9 does not have an IgM-specific mode of inhibition.

Major Point 6) The reviewer makes several related points regarding the observed calcium-specific nature of Gal-9 suppression in B cells:

a) *“Fig. 5 and Fig. S4: The authors should discuss possible reasons why Gal-9 inhibits BCR-induced Ca²⁺ signaling but not other aspects of BCR signaling.”*

Our Response. As suggested, we have expanded this section of the Discussion to consider how Gal-9 may uniquely regulate calcium signaling, but not other signaling pathways capable of being targeted by SHP-1. There are several possible explanations for this observation:

- i. Gal-9 may specifically lead to targeting of SHP-1 to plasma membrane calcium efflux pumps, such as PMCA4a, as has been previously described for CD22 and SHP-1¹¹.
- ii. Lyn has been shown to be a dual regulator of BCR signaling, promoting positive signaling via phosphorylation of both ITAM-containing proteins (such as CD79) and ITIM-containing proteins (such as CD22)¹². Thus, although SHP-1 may indeed be targeting critical BCR signaling components, these effects may be counterbalanced by the simultaneous positive signaling induced by Lyn. Therefore, the “net” phosphorylation state may be unchanged for BCR signaling molecules such as BLNK, Syk, and PLC γ II, but favored toward activation for calcium efflux pumps such as PMCA4a.
- iii. Although we observed only one major band in our Gal-9 immunoprecipitation, we cannot exclude the possibility that Gal-9 may also interact with other glycoprotein receptors, including those that positively regulate BCR signaling. Again, in this case, the observed phosphorylation status of each BCR signaling effector may represent the net input from several Gal-9-binding glycoproteins (or glycolipids).

b.) The reviewer recommends that, “*The CD22/Lyn/SHP-1 pathway has been shown to inhibit BCR-induced phosphorylation of BLNK, BTK, and PLC-gamma, which constitute a central Ca²⁺ signaling module. Showing that Gal-9 inhibited the phosphorylation of these proteins would provide more mechanistic insights and argue against alternative explanations, e.g. Gal-9 affecting the function of plasma membrane Ca²⁺ channels.*”

Our Response. We agree with the reviewer and were surprised to find the majority of proximal BCR signaling components to be unaffected (or even enhanced) by Gal-9, including BLNK and PLC γ II (**Supplementary Fig. 6c,d**). From these data, we therefore conclude that the most likely mechanism of action of Gal-9 is not suppression of canonical BCR signaling modules, but an alternative mechanism such as activation of calcium efflux pumps. This is also supported by the ability of Gal-9 to affect ionomycin-induced calcium flux (see Figure corresponding with **Major Point 6e**). The precise mechanism of action of Gal-9 will be an important area of follow-up study by our laboratory.

c.) The reviewer recommends that, “*Alternatively, anti-IgM concentrations required for BCR-induced Ca²⁺ signaling may be higher than for other signaling responses, and hence more sensitive to modest inhibition of proximal BCR signaling events. Experiments using sub-optimal (i.e. less than saturating) concentrations of anti-IgM may reveal that other BCR signaling pathways are also modulated by Gal-9. This type of experiment would fit with the idea that Gal-9 tunes the threshold for B cell activation.*”

Our Response. We thank the reviewer for this suggestion. In our hands, high concentrations of BCR crosslinking antibody (>10 μ g/mL) were necessary to obtain measurable signaling responses in Western blot and calcium assays; however, we acknowledge that, under these stimulation conditions, we cannot formally rule out the possibility that finer effects of Gal-9 on BCR signaling are obscured at these concentrations. We have now added additional commentary in the Discussion to point out this limitation in our approach.

d.) The reviewer recommends that, “*In Fig. S4, Gal-9 alone causes a 4- to 10- increase in pAkt and pERK that is reversible by lactose. The authors should discuss whether this represents an increase in basal BCR signaling caused by Gal-9.*”

Our Response. We indeed observe consistent induction of pAkt and pErk when Gal-9 is administered in the absence of BCR stimulus. Additionally, several BCR-related signaling molecules (e.g. Syk, BLNK) also trend toward increased phosphorylation status in response to Gal-9.

We propose two possible explanations for this observation:

- i. The findings reflect the dual roles of Lyn in BCR signaling, where Lyn can either act to positively or negatively regulate BCR signaling depending on the degree of stimulus¹². The level of Lyn phosphorylation observed in response to Gal-9 treatment in the absence of BCR crosslinking may slightly favor Lyn phosphorylation of positive regulators, such as Syk. By contrast, additional Lyn phosphorylation induced by Gal-9 under conditions of strong Lyn phosphorylation (i.e. BCR crosslinking) may tilt the balance toward acting on negative regulators, such as CD22 and consequently SHP-1.
- ii. Under certain circumstances, CD22 may have positive signaling roles. Consistent with this idea, the cytoplasmic tail of CD22 contains docking sites for Syk and PLC γ ¹³. In the absence of BCR crosslinking, phosphorylation of CD22 could theoretically be driving the docking and activation of Syk to the CD22 cytoplasmic tail.
- iii. Positive BCR signaling may be occurring through the action of a separate Gal-9 binding glycoprotein.

We have modified the Discussion accordingly to address these possibilities.

e.) The reviewer recommends that, "*In Fig. 5f it would be good to add a control to show that exogenous Gal-9 does not inhibit ionomycin-induced nuclear translocation of NFAT1.*"

Our Response. We agree with the reviewer that this would be an interesting experiment. While we did not directly examine whether Gal-9 could influence ionomycin induced nuclear translocation of NFAT1, we did test whether Gal-9 could dampen Ca²⁺ flux caused by ionomycin. Interestingly, we found that Gal-9 could in fact suppress ionomycin-induced calcium mobilization (**below**), supporting the hypothesis that Gal-9 acts via a mechanism involving calcium efflux pumps (such as the SHP-1 target PMCA4a) rather than a BCR-specific mechanism.

Figure corresponding with Reviewer 3 Major Point 6e

Galectin-9 suppresses ionomycin-induced calcium flux. Calcium flux curves of CD19+ CD44+ tonsillar B cells loaded with Fluo-4 calcium indicator dye and treated with 0.1 μ M ionomycin, with or without 2.5 μ g/mL recombinant human Gal-9. Mean fluorescence intensities (MFI) over the entire acquisition period were normalized to the average MFI of the 30s baseline. *Arrow*, time of stimulus addition. For complete description of calcium flux protocol, see **Fig. 5a** and **Materials and Methods**.

f.) The reviewer recommends that, “Fig. 5d-f. Immunofluorescence (IF) is a much better technique for quantifying nuclear translocation than cell fractionation, which has many caveats. The IF data in Fig. 5f should be quantified as percent of cells with NFAT in the nucleus.”

Our Response. We concur with the reviewer and have now added appropriate quantification and statistical calculations (**Fig. 5g**).

Major Point 7) The reviewer makes the following comment: “Fig. S5. The authors show that treating B cells with anti-IgM + Gal-9 causes co-capping of IgM, CD45, and pCD22. From these data, they argue that this supports a model in which CD45 brings CD22 close to the BCR and thereby inhibits BCR signaling. The physiological relevance of “capping” is not clear ... As well, controls are needed to show that Gal-9 doesn’t cause other membrane proteins to cap. [Additionally], These images are of low magnification and resolution and do not support the authors’ proposal that Gal-9 promotes the co-localization of CD22 and the BCR in Lyn-containing signaling domains (Fig. 7). Higher magnification microscopy images as well as higher resolution techniques such as proximity-ligation assays or super-resolution microscopy would be needed to support this model.”

Our Response. We shared the reviewer’s curiosity about whether capping by Gal-9 was specific to CD45 and CD22. We therefore performed IF imaging of CD20 following Gal-9 treatment. While it was clear that CD20 did not cap to the same extent as CD45, we did observe a partial capping response that was reminiscent of that of pCD22 (data not shown). This result suggests that our interpretation of a specific pCD22 “capping” effect in this experiment may be misguided. Therefore, we have removed this panel from the paper and have modified our model (**Fig. 8**) accordingly to be less speculative regarding Gal-9-induced alterations in membrane localization. However, while we cannot draw conclusions regarding CD45 / CD22 / IgM co-localization, we now present new evidence that the effects of Gal-9 on BCR calcium signaling are indeed dependent on CD22 and SHP-1 (see **Major Point 3**).

Major Point 8) The reviewer makes the following recommendation: “Fig. 6. The experimental results using Gal-9 +/- lactose could be strengthened by repeating a few key signaling experiments in the paired cell lines with and without GCB-like I-branched membrane proteins. Because the cell lines are identical except for GCNT2 expression, this would reveal whether I-branching affects BCR signaling by itself and be an excellent direct test of whether I-branching renders B cells resistant to the effects of Gal-9.”

Our Response. We agree with the reviewer that the GCNT2 variant cell lines would theoretically be an excellent model to directly test whether I-branches protect against the effects of Gal-9. However, we noted several differences between healthy primary cells and B lymphoma cell lines that has limited their utility in functional assays. For example, we observed that many B lymphoma cell lines exhibit reduced levels of CD22, SHP-1, and/or Gal-9, including Ramos B cells (SHP-1⁻ Gal-9⁻) and NUDUL-1 B cells (CD22^{low} SHP-1⁻), complicating functional experiments involving GCNT2KD / OE variants (**Fig. 7b**).

Nevertheless, in effort to find a suitable cell line model for functional experiments, we identified Karpas 1718 B cells, which express Gal-9, CD22, and SHP-1, albeit at much lower levels than primary cells for the latter two (**Fig. 7b**). This cell line was particularly useful for experiments addressing the effects of knocking down endogenous Gal-9 (see **Major Points 1 and 9** and **Fig. 7c-f**). Unfortunately,

however, though overexpression of GCNT2 in this cell line led to a substantial reduction in Gal-9 binding (~10-fold), we found that Gal-9 binding was still apparent even in the presence of N-glycan I-branches (see **Figure panel “a”** on next page), particularly when the cells were not washed following Gal-9 treatment (such as during calcium assays). These data lead us to speculate that Karpas 1718 B cells also express poly-LacNAc (Gal-9 binding determinants) on O-linked glycans, which would not likely be modified by GCNT2 due to its reported preference for N-glycan polyLacNAcs^{14,15}. This reasoning is supported by high expression by Karpas 1718 B cells of GCNT1, which catalyzes poly-LacNAc synthesis on Core 2 O-glycans (see **Figure panel “b”** on next page) and promotes galectin-3 and galectin-9 binding¹⁶⁻¹⁸. Given that CD45 on B cells is a mucinous heavily O-glycosylated molecule, it stands to reason that GCNT2 in isolation may be insufficient to fully block Gal-9 binding in the presence of O-glycan poly lactosamines.

By contrast, on primary cells, we suspect that the loss of Gal-9 activity in GC B cells results from a combination of factors, including: (1) upregulation of GCNT2 / I-branches to reduce Gal-9 binding to N-glycans (**Fig. 1** and **Fig. 2**); (2) decreased Core 2 O-glycan poly lactosamine via GCNT1 downregulation (manuscript in preparation); (3) downregulation of Gal-9 itself (**Fig. 3b,c**). Therefore, we speculate that parsing the specific role of GCNT2 in B cell lines will require more sophisticated approaches that manipulate GCNT2 expression in concert with Gal-9 and GCNT1.

Figure corresponding with Reviewer 3 Major Point 8

Analysis of Gal-9 binding to GCNT2OE Karpas 1718 B cells and GCNT1 expression between NUDUL-1 and Karpas 1718 B cells. (a) Karpas 1718 control overexpression (GFP) or GCNT2 overexpression (GCNT2OE) B cells were generated by lentiviral transduction, and binding of recombinant Gal-9 was assessed as in **Fig. 2d**. **(b)** Comparison of relative transcript expression of the Core 2 N-acetylglucosaminyl transferase, *GCNT1*, between NUDUL-1 B cells and Karpas 1718 B cells. Data are from the publicly available dataset, GSE42204. More complete information about experimental protocols can be found in **Materials and Methods**.

Major Point 9) The reviewer makes the following recommendation: “*Fig. 7. No functional evidence is presented to support the idea that Gal-9 inhibits BCR signaling largely by activating via CD22/Lyn/SHP1 pathway. This should be tested by using siRNA or shRNA (as in Fig. 6) to knock down the expression of CD22 or SHP-1 in Ramos cells and seeing if this largely ablates the inhibitory effects of Gal-9 on BCR signaling. This is a critical experiment to support the authors’ signaling data and model.*”

Our Response. We agree that additional experiments are warranted to support a CD22- and SHP-1-dependent mechanism of action of Gal-9 in B cells. We now present two new lines of evidence to support our hypothesis of CD22 / SHP-1 dependency:

- i. **Gal-9 knockdown in CD22+ / SHP-1+ B cells, but not B cells lacking CD22 and SHP-1, leads to enhanced BCR-induced intracellular calcium levels.** To test whether CD22 and SHP-1 are required for Gal-9’s activity in B cells, we took advantage of the observed natural heterogeneity in CD22 and SHP-1 expression across B cell lines, specifically between two Gal-9^{hi} cell lines that differ in expression of CD22 and SHP-1: Karpas 1718 cells, which express moderate levels of CD22⁺ and SHP-1⁺, and NUDUL-1 B cells, which lack meaningful expression of CD22 and SHP-1 (**Fig. 7b**). We therefore reasoned that knockdown of Gal-9 in these two cell lines would provide a natural experimental setting to test whether CD22 and SHP-1 are required for Gal-9 suppressive activity. Strikingly, though similar levels of knockdown were achieved in both cell lines (**Supplementary Figure 8b,c**), only knockdown of Gal-9 in the CD22⁺ SHP-1⁺ cell line (Karpas 1718) resulted in enhanced peak and total calcium mobilization in response to anti-BCR stimulus, whereas Gal-9 knockdown in CD22^{lo} SHP-1^{lo} B cells (NUDUL-1) had no discernable effect (**Fig. 7c-j**).
- ii. **CD22 / CD45 dissociation by sialidase treatment prevents inhibitory effect of Gal-9 on B cell receptor calcium signaling.** CD45 and CD22 have been reported to associate in B cells at steady state via interactions between the sialic acid binding domain of CD22 and the sialylated ectodomain of CD45^{3,4,7}. Moreover, previous studies have utilized bacterial sialidase treatment of whole cells to disengage CD22 from its ligands *in cis*, including CD45^{3,6,7} (**Supplementary Fig. 7d**). As expected from previous reports on sialidase treatment of B cells, enzymatic removal of sialic acids led to universally dampened calcium flux, likely due to activation of CD22 by “unmasking” it from its ligands *in cis* (termed “unmasking”)^{6,7} (**Fig. 5h**, left panel). Nonetheless, BCR calcium flux was still apparent following BCR crosslinking of sialidase-treated cells, allowing us to test whether Gal-9 inhibits calcium flux via the same, or different, pathway as CD22. In support of our hypothesis, Gal-9 was completely unable to suppress BCR-mediated calcium mobilization in sialidase treated (CD22 / CD45 decoupled) B cells (**Fig. 5h**, middle and right panels). Importantly, this effect was not due to altered Gal-9 binding activity, as Gal-9 bound equally to sialidase treated and untreated cells (**Supplementary Fig. 7e**). Therefore, taken together with experiments assessing Gal-9KD in CD22/SHP-1 sufficient / deficient cell lines, these data support the hypothesis that Gal-9 inhibits calcium flux via the CD22-SHP-1 axis.

Minor Point 1) The reviewer makes the following recommendation: “*The legend to Fig. 1a should indicate what each of the colors represents – naïve, GC, and memory B cells?*”

Our Response. We thank the reviewer for this suggestion. We have now added labels to **Fig. 1a** to more easily identify what each color represents.

Minor Point 2) The reviewer makes the following recommendation: “*Figs. 1c, 1e and Fig. S1. For the glycome analysis, the figure legends should provide information about reproducibility, specifically how many times the same analysis was performed on samples from different individuals, and what the variability between samples was.*”

Our Response. We agree with the reviewer and now have added the appropriate information in the legend of **Fig. 1**. The results in **Fig. 1** are from one tonsil specimen and are representative of one of two experiments using different tonsil specimens. Due to the resource intensive nature of the experiment (>40 million cells for each condition), the second experiment was restricted to analysis of naïve and GC B cells. Acknowledging the limited statistical power of mass spectrometric analyses, we also performed plant lectin-based flow cytometry in tandem to further reinforce MS results and confirm the I-branching phenotype in numerous tonsil specimens (n=10), with appropriate statistical power (**Supplementary Fig. 3**). Taken together, we believe that this approach provides the appropriate sensitivity to detect relevant glycan features in an unbiased fashion (via MS analysis), while allowing for more exhaustive, confirmatory analysis of specific glycan features across numerous tonsil specimens (via plant lectins analysis).

Minor Point 3) The reviewer makes the following recommendation: “*Fig. 4a should be moved to supplemental data as the identification of these 4G10-reactive bands is not definitive and Fig. 4b directly tests the hypothesis that they are the phosphorylated forms of CD22 and Lyn.*”

Our Response. We thank the reviewer for this suggestion, and have modified the figures accordingly (**Fig. 4** and **Supplementary Fig. 6**).

Minor Point 4) The reviewer makes the following recommendation: “*There is a mistake in legend to Fig. 5. CsA and iono were used in Fig. 5d, but 3 lines from the end of the legend it says CsA and PMA (instead of iono).*”

Our Response. We thank the reviewer and apologize for this discrepancy. The figure legend now correctly describes the experimental protocol as using ionomycin.

References:

- 1 Nabi, I. R., Shankar, J. & Dennis, J. W. The galectin lattice at a glance. *J Cell Sci* **128**, 2213-2219, doi:10.1242/jcs.151159 (2015).
- 2 Treanor, B. & Batista, F. D. Organisation and dynamics of antigen receptors: implications for lymphocyte signalling. *Curr Opin Immunol* **22**, 299-307, doi:10.1016/j.coi.2010.03.009 (2010).
- 3 Gasparrini, F. *et al.* Nanoscale organization and dynamics of the siglec CD22 cooperate with the cytoskeleton in restraining BCR signalling. *EMBO J* **35**, 258-280, doi:10.15252/embj.201593027 (2016).
- 4 Coughlin, S. *et al.* An extracatalytic function of CD45 in B cells is mediated by CD22. *Proc Natl Acad Sci U S A* **112**, E6515-6524, doi:10.1073/pnas.1519925112 (2015).
- 5 Chen, I. J., Chen, H. L. & Demetriou, M. Lateral compartmentalization of T cell receptor versus CD45 by galectin-N-glycan binding and microfilaments coordinate basal and activation signaling. *J Biol Chem* **282**, 35361-35372, doi:10.1074/jbc.M706923200 (2007).
- 6 Razi, N. & Varki, A. Masking and unmasking of the sialic acid-binding lectin activity of CD22 (Siglec-2) on B lymphocytes. *Proc Natl Acad Sci U S A* **95**, 7469-7474 (1998).
- 7 Mahajan, V. S. & Pillai, S. Sialic acids and autoimmune disease. *Immunol Rev* **269**, 145-161, doi:10.1111/imr.12344 (2016).
- 8 Bi, S., Hong, P. W., Lee, B. & Baum, L. G. Galectin-9 binding to cell surface protein disulfide isomerase regulates the redox environment to enhance T-cell migration and HIV entry. *Proc Natl Acad Sci U S A* **108**, 10650-10655, doi:10.1073/pnas.1017954108 (2011).
- 9 Lu, L. H. *et al.* Characterization of galectin-9-induced death of Jurkat T cells. *J Biochem* **141**, 157-172, doi:10.1093/jb/mvm019 (2007).
- 10 Liu, W., Sohn, H. W., Tolar, P. & Pierce, S. K. It's all about change: the antigen-driven initiation of B-cell receptor signaling. *Cold Spring Harb Perspect Biol* **2**, a002295, doi:10.1101/cshperspect.a002295 (2010).
- 11 Chen, J. *et al.* CD22 attenuates calcium signaling by potentiating plasma membrane calcium-ATPase activity. *Nat Immunol* **5**, 651-657, doi:10.1038/ni1072 (2004).
- 12 Xu, Y., Harder, K. W., Huntington, N. D., Hibbs, M. L. & Tarlinton, D. M. Lyn tyrosine kinase: accentuating the positive and the negative. *Immunity* **22**, 9-18, doi:10.1016/j.immuni.2004.12.004 (2005).
- 13 Law, C. L. *et al.* CD22 associates with protein tyrosine phosphatase 1C, Syk, and phospholipase C-gamma(1) upon B cell activation. *J Exp Med* **183**, 547-560 (1996).
- 14 Renkonen, O. Enzymatic in vitro synthesis of I-branches of mammalian polylectosamines: generation of scaffolds for multiple selectin-binding saccharide determinants. *Cell Mol Life Sci* **57**, 1423-1439, doi:10.1007/PL00000627 (2000).
- 15 Stone, E. L. *et al.* Glycosyltransferase function in core 2-type protein O glycosylation. *Mol Cell Biol* **29**, 3770-3782, doi:10.1128/MCB.00204-09 (2009).
- 16 Clark, M. C. *et al.* Galectin-3 binds to CD45 on diffuse large B-cell lymphoma cells to regulate susceptibility to cell death. *Blood* **120**, 4635-4644, doi:10.1182/blood-2012-06-438234 (2012).
- 17 Schaefer, K. *et al.* Galectin-9 binds to O-glycans on protein disulfide isomerase. *Glycobiology* **27**, 878-887, doi:10.1093/glycob/cwx065 (2017).
- 18 Thiemann, S. & Baum, L. G. Galectins and Immune Responses-Just How Do They Do Those Things They Do? *Annu Rev Immunol* **34**, 243-264, doi:10.1146/annurev-immunol-041015-055402 (2016).

REVIEWERS' COMMENTS:

Reviewer #1 (Remarks to the Author):

The authors have adequately addressed my concerns.

Reviewer #3 (Remarks to the Author):

The authors have done an outstanding job responding to the suggestions made by all 3 reviewers, many of which overlapped. A number of new experiments have been added to clarify and extend key points. In addition, the summary figure is now much less speculative and is well supported by the novel and interesting data presented in this manuscript. The authors should be commended for their detailed and thoughtful responses to the reviewers' comments.

POINT-BY-POINT RESPONSES TO REVIEWERS

Reviewer #1

The reviewer states, "*The authors have adequately addressed my concerns.*"

Our Response: Thank you! We acknowledge that all changes are satisfactorily addressed.

Reviewer #3

The reviewer states, "*The authors have done an outstanding job responding to the suggestions made by all 3 reviewers, many of which overlapped. A number of new experiments have been added to clarify and extend key points. In addition, the summary figure is now much less speculative and is well supported by the novel and interesting data presented in this manuscript. The authors should be commended for their detailed and thoughtful responses to the reviewers' comments.*"

Our Response: Thank you! We acknowledge that all changes are satisfactorily addressed.